# 4-(Phenylselanyl)-2H-chromen-2-one-Loaded Nanocapsule Suspension—A Promising Breakthrough in Pain Management: Comprehensive Molecular Docking, Formulation Design, and Toxicological and Pharmacological Assessments in Mice

**DOI:** 10.3390/pharmaceutics16020269

**Published:** 2024-02-14

**Authors:** Caren Aline Ramson da Fonseca, Vinicius Costa Prado, Jaini Janke Paltian, Jean Carlo Kazmierczak, Ricardo Frederico Schumacher, Marcel Henrique Marcondes Sari, Larissa Marafiga Cordeiro, Aline Franzen da Silva, Felix Alexandre Antunes Soares, Robson da Silva Oliboni, Cristiane Luchese, Letícia Cruz, Ethel Antunes Wilhelm

**Affiliations:** 1Graduate Program in Biochemistry and Bioprospecting, Biochemical Pharmacology Research Laboratory, Federal University of Pelotas, Pelotas CEP 96010-900, RS, Brazil; carenramson@hotmail.com (C.A.R.d.F.); jaini_paltian@hotmail.com (J.J.P.); cristiane_luchese@yahoo.com.br (C.L.); 2Graduate Program in Pharmaceutical Sciences, Pharmaceutical Technology Laboratory, Federal University of Santa Maria, Santa Maria CEP 97105-900, RS, Brazil; vini132007@gmail.com; 3Graduate Program in Chemistry, Chemistry Department, Federal University of Santa Maria, Santa Maria CEP 97105-900, RS, Brazil; jeanckazmierczak@gmail.com (J.C.K.); ricardo.schumacher@ufsm.br (R.F.S.); 4Clinical Analysis Department, Federal University of Paraná, Curitiba CEP 80210-170, PR, Brazil; marcelsarih@hotmail.com; 5Graduate Program in Biological Sciences: Toxicological Biochemistry, Federal University of Santa Maria, Santa Maria CEP 97105-900, RS, Brazil; larissa.marafiga@hotmail.com (L.M.C.); alinefranzen@hotmail.com (A.F.d.S.); felix@ufsm.br (F.A.A.S.); 6Center for Chemical, Pharmaceutical, and Food Sciences, CCQFA, Federal University of Pelotas, Pelotas CEP 96010-900, RS, Brazil; rooliboni@gmail.com

**Keywords:** nanotechnology, pain, *Caenorhabditis elegans*, mice, organoselenium, coumarin

## Abstract

Therapies for the treatment of pain and inflammation continue to pose a global challenge, emphasizing the significant impact of pain on patients’ quality of life. Therefore, this study aimed to investigate the effects of 4-(Phenylselanyl)-2H-chromen-2-one (4-PSCO) on pain-associated proteins through computational molecular docking tests. A new pharmaceutical formulation based on polymeric nanocapsules was developed and characterized. The potential toxicity of 4-PSCO was assessed using *Caenorhabditis elegans* and *Swiss* mice, and its pharmacological actions through acute nociception and inflammation tests were also assessed. Our results demonstrated that 4-PSCO, in its free form, exhibited high affinity for the selected receptors, including p38 MAP kinase, peptidyl arginine deiminase type 4, phosphoinositide 3-kinase, Janus kinase 2, toll-like receptor 4, and nuclear factor-kappa β. Both free and nanoencapsulated 4-PSCO showed no toxicity in nematodes and mice. Parameters related to oxidative stress and plasma markers showed no significant change. Both treatments demonstrated antinociceptive and anti-edematogenic effects in the glutamate and hot plate tests. The nanoencapsulated form exhibited a more prolonged effect, reducing mechanical hypersensitivity in an inflammatory pain model. These findings underscore the promising potential of 4-PSCO as an alternative for the development of more effective and safer drugs for the treatment of pain and inflammation.

## 1. Introduction

Pain is a debilitating symptom that can impact a person’s well-being and quality of life [1]. Acute pain arises from tissue trauma and inflammation, while chronic pain can lead to debilitating diseases [2,3,4]. While various treatments exist for acute pain, the condition’s etiology is intricate, encompassing sensory, affective, cognitive, and behavioral dimensions [5,6]. Traditional pain relief methods are often inadequate, requiring drug combinations like opioids and anti-inflammatory drugs for effective relief [6,7].

Considering this, extensive research on organoselenium compounds has unfolded, revealing diverse applications and promising effects in preventing oxidative stress and inflammatory conditions over the years [8]. However, it is known that organic selenium compounds have numerous pharmacological targets, and, for this reason, are capable of causing a complex pattern of action, and this characteristic can lead to molecular toxicity [9]. In this way, coumarins, regarded as privileged structures in medicinal chemistry, offer straightforward synthesis and versatile functionalization within the coumarin ring, yielding derivatives with compelling applications and biological activities [10]. These compounds exhibit notable antioxidant, anti-inflammatory, antitumoral, antimicrobial, and antiviral actions, supported by a body of literature [8,11,12,13]. 

In this context, there has been a growing interest in investigating seleno–coumarin compounds due to their unique chemical structure. The combination of selenium organic compounds with coumarin derivatives allows selenocoumarinic compounds to exhibit promising pharmacological effects, as evidenced by preclinical studies. Notably, research by Padilha et al. [14] showcased the antioxidant activity of seleno–coumarin compounds, suggesting their potential in reducing oxidative stress associated with pain and inflammation. Furthermore, studies conducted by Lagunes et al. [15] revealed the antiproliferative activity of seleno–coumarins in tumor cell lines, highlighting their potential to minimize side effects linked to cancer treatment. The works of Arsenyan et al. [16] and Domracheva et al. [17] reported on the antioxidant and anticancer activities of seleno–coumarin isomers. Most recently, Yildirim et al. [18] delved into the pharmacological effects of 3-acetyl coumarin–selenophene in DU-145 tumor cells, exploring its activities during apoptosis and oxidative stress. These findings further boost the study and investigation of the pharmacological properties of 4-(Phenylselanyl)-2H-chromen-2-one (4-PSCO), an innovative and promising compound belonging to this class of compounds on the rise.

In light of these considerations, alternative approaches remain pertinent in enhancing patients’ quality of life, particularly in pain relief. Despite advancements in pharmaceutical research, validating the pharmacological effects of synthetic compounds remains a persistent challenge, necessitating careful consideration of factors such as absorption, metabolism, distribution, elimination, and potential toxicity, which can significantly impact treatment efficacy [19]. In this regard, nanocarrier systems, particularly polymeric nanostructures, emerge as promising tools for optimizing drug properties with minimal health or environmental repercussions [20,21,22]. These systems offer potential solutions to challenges in pharmacokinetics, pharmacodynamics, bioavailability augmentation, and tissue or cellular specificity [23,24]. Therefore, we believe that the combination of organoselenium compounds and nanocarriers constitutes an alternative that has already been explored and demonstrates significant advantages. Several studies have been conducted, and the results are promising, especially for pain treatment [19,21]. 

In alignment with the potential demonstrated by organic selenium compounds and coumarin derivatives, coupled with the promising outlook of 4-PSCO in both its free and nanoencapsulated forms as a pivotal contender in pain and inflammation therapeutics, we propose a comprehensive study. Due to a lack of knowledge regarding compounds similar to seleno–coumarins used in pain and inflammation treatment, this study aims to investigate the effects of 4-PSCO on pain-related proteins using computational molecular docking tests. Additionally, it involves the preparation and characterization of 4-PSCO polymeric nanocapsules. The scope encompasses assessments of formulation stability, toxicological potential, and acute antinociceptive effects, thereby contributing to an enriched understanding of the compound’s multifaceted pharmacological profile and safety.

## 2. Materials and Methods

### 2.1. Animals and Ethical Approval

Sixty-day-old male and female adult *Swiss* mice were obtained from the Federal University of Pelotas, Brazil. The animals were kept in a separate animal room, in a 12 h light/dark cycle (with lights on at 6:00 a.m.), at a room temperature of 22 ± 2 °C, with free access to food and water. The animal care and behavioral assays were conducted in compliance with the National Institutes of Health Guide for the Care and Use of Laboratory Animals (NIH Publications No. 823, revised 1978) and International Guiding Principles for Biomedical Research Involving Animals. The experimental protocols were authorized by the Committee on Care and Use of Experimental Resources of the Federal University of Pelotas (Brazil), affiliated with the National Council for the Control of Animal Experimentation, and registered under the number CEEA 13049-2021. All efforts were made to minimize the number of animals used and their suffering.

### 2.2. Drugs and Reagents

Ethylcellulose polymer (Ethocel^®^TM Standard 10 Premium) was generously provided by Colorcon (São Paulo, Brazil). Polysorbate 80 (Tween^®^ 80) and medium-chain triglycerides (MCT) were supplied by Delaware (Porto Alegre, Brazil), while sorbitan monooleate (Span^®^ 80) was sourced from Sigma Aldrich (São Paulo, Brazil). Acetone was obtained from CRQ Quimica (São Paulo, Brazil), and high-performance-liquid-chromatography-grade acetonitrile was acquired from Li-Chrosolv (São Paulo, Brazil). All remaining reagents and solvents were of analytical grade and were utilized without further modification. The synthesis and characterization of 4-PSCO (Figure 1) were conducted at the Department of Chemistry at the Federal University of Santa Maria [14]. Analysis of GC/MS determined the chemical purity of this compound (99.9%). The 4-PSCO was dissolved in canola oil. Morphine and meloxicam (used as reference drugs) were dissolved in an isotonic saline solution. Meloxicam and morphine (Sigma Aldrich, São Paulo, Brazil) were administered intragastrically (i.g.) and intraperitoneally (i.p.), respectively, at a constant volume of 10 mL/kg of body weight.

### 2.3. Computational Methods

An initial conformation of the compound 4-PSCO was obtained using CREST [25] at the GFN2-xtb level of theory [26], employing water as the solvent (Figure 2). Docking simulations were performed with the following target proteins: p38 mitogen-activated protein kinase (MAPK, PDB: 3Z1A) [27], Janus kinase 2 (JAK, PDB: 2B7A) [28], toll-like receptor 4 (TLR4, PDB: 3FXI) [29], nuclear factor-kappa β (NFkB, PDB: 1NFK) [30], peptidyl arginine deiminase type 4 (PAD4, PDB: 1WDA) [31], and phosphoinositide 3-kinase γ isoform (PI3K, PDB: 3DPD) [32]. The crystal structures of the receptors were obtained from the Protein Data Bank and prepared using Autodock tools [33]. Protein–ligand interactions were described using Discovery Studio 2021 [34]. All docking calculations were performed using AutoDock Vina [35]. Electronic structure calculations were carried out with ORCA 4.2.1 [36] using the B3LYP functional [37,38] and def2-TZVP basis set [39], incorporating the D3BJ dispersion correction [40,41]. The interaction between the selected proteins and the reference drugs commonly used to treat pain and inflammation (ketoprofen, naproxen, tofacitinib, dexamethasone, and methotrexate) was also evaluated.

### 2.4. Analytical Method for the 4-PSCO Quantification

The quantification of 4-PSCO was performed using the high-performance liquid chromatography method with UV detection (HPLC-UV) on a Shimadzu LC-10A system from Japan. The system included an LC-20AT pump, a UV–Vis SPD-M20A detector, a CBM-20A system controller, and an SIL-20A HT autosampler valve. A C18 column (Agilent, reversed phase, 5 μm, 110 Å, 150 mm × 4.60 mm) was employed. The mobile phase consisted of an isocratic system (30% ultrapure water and 70% acetonitrile) at a flow rate of 1.0 mL/min. The injection volume was set at 20 μL, and the quantification of 4-PSCO in samples was conducted at 213 nm with a retention time of 5.3 min. The method for quantifying 4-PSCO was validated following the International Conference on Harmonization (ICH) guidelines, demonstrating linearity (r = 0.99), specificity (101.8 to 100.3%), and precision (relative standard deviation ≤ 0.68%) within the concentration range of 5.0–25.0 μg/mL.

### 2.5. 4-PSCO-Loaded Polymeric Nanocapsule Suspension Preparation

The 4-PSCO-loaded polymeric nanocapsule suspensions (4-PSCO NC) were prepared (3 batches) by interfacial deposition of the preformed polymer method [42]. Initially, an organic phase comprising 4-PSCO (10 mg), ethylcellulose polymer (100 mg), MCT oil (300 mg), Span^®^ 80 (77 mg), and acetone (27 mL) underwent moderate magnetic stirring at 40 °C for 1 h until complete dissolution of the components. Subsequently, this phase was injected into an aqueous dispersion of Tween^®^ 80 (77 mg in 53 mL) and maintained under magnetic stirring at room temperature. After 10 min, acetone was removed, and the resulting suspension was concentrated under reduced pressure (Rotary evaporator R 114, Büchi^®^) until reaching a final volume of 10 mL, corresponding to a 4-PSCO concentration of 1 mg/mL (4-PSCO NC). Placebo nanocapsules (NC P) were also prepared using a similar procedure but without the addition of 4-PSCO [42].

### 2.6. 4-PSCO-Loaded Polymeric Nanocapsule Suspension Physicochemical Characterization

The average particle size and polydispersity index (PDI) were determined through dynamic light scattering (DLS) using the Zetasizer^®^ Nano-ZS ZEN 3600 model from Malvern Instruments, Malvern, UK. After diluting the suspensions (20 µL) in previously filtered ultrapure water (10 mL, 0.45 µm Millipore^®^, Burlington, MA, USA), measurements were taken. Additionally, the volume-weighted mean diameters [D4;3] and polydispersity (SPAN) were assessed using laser diffraction (Mastersizer 3000, Malvern Instruments, UK). The sample was directly introduced into the disperser compartment of the equipment, containing 250 mL of distilled water, until the appropriate obscuration index was achieved (10–12%). The refractive index of the ethylcellulose polymer (1.47) was utilized in the analysis. Subsequently, the volume-weighted mean diameters [D4;3] were determined and expressed in µm for each nanocapsule suspension batch. SPAN values, used as indicators of the polydispersity of formulations, were computed using Equation (1).
(1)$$\mathrm{SPAN}=\frac{\left(Dv\left(90\right)-Dv\left(50\right)\right)}{Dv\left(10\right)}$$
where Dv (90), Dv (10), and Dv (50) are the diameters at 90%, 10%, and 50% of the cumulative distribution of the diameter curve, respectively.

The zeta potential was determined by electrophoretic mobility (Zetasizer^®^ Nano-ZS ZEN 3600 model, Malvern Instruments, UK). Samples (20 µL) were diluted in 10 mM (10 mL) NaCl solution, previously filtered (0.45 µm, Millipore^®^). The pH values were determined using a calibrated potentiometer (Simpla Instruments, Brazil), which was directly immersed in the formulations [42]. 

The total 4-PSCO content was determined after its extraction from the polymeric nanocapsule suspensions (150 μL) using methanol (10 mL). Subsequently, the solution underwent sonication using the ultrasonic bath Q3.0/40A model from Ultronique, Brazil, for a duration of 10 min, followed by filtration through a 0.45 µm Millipore^®^ filter. The filtered solution was then subjected to analysis using the HPLC method (Section 2.4). For the determination of encapsulation efficiency (EE %), a 300 µL aliquot of the nanocapsule suspensions was transferred to a 10.000 MW centrifugal device (Amicon^®^ Ultra, Millipore), and free 4-PSCO was separated using the ultrafiltration technique (20 min at 2.200 xg). The encapsulation efficiency percentage (EE%) was calculated by subtracting the concentration of free 4-PSCO in the ultrafiltrate from the total concentration of 4-PSCO in the nanocapsules, as per Equation (2): (2)$$EE\;\%=\frac{Total\;4-PSCO\;content-Free\;4-PSCO\;content}{Total\;4-PSCO\;content}\times 100$$

The 4-PSCO NC stability was monitored for 15 and 30 days after preparation by determining the compound content, pH, particle size, PDI, and zeta potential. Formulations were stored in glass flasks at room temperature (25 ± 2 °C) [42].

### 2.7. In Vitro Release of 4-PSCO from Polymeric Nanocapsules

The release profile of 4-PSCO from polymeric nanocapsules was performed in triplicate by the dialysis bag diffusion method [43]. An aliquot of 1 mL of 4-PSCO-loaded polymeric nanocapsules was placed inside the dialysis cellulose membrane (molecular weight 10,000 Da, Sigma-Aldrich, St. Louis, MI, USA) and immersed in 200 mL of release medium composed of 70% of the 1 M potassium phosphate buffer at pH 7.4 and 30% of ethanol kept under moderate magnetic stirring at 37 °C. At predetermined times (1, 2, 3, 4, 5, 6, 8, 10, 12, 24, and 48 h), 1 mL of the external medium was collected, and the same volume of fresh medium was replaced. The quantity of 4-PSCO released was assessed by the HPLC method previously described in Section 2.4. A methanolic solution of 4-PSCO (1 mg/mL) was simultaneously evaluated in triplicate for comparison purposes. The results were expressed as % of 4-PSCO released.

### 2.8. Evaluation of Toxicity in a Model of Caenorhabditis elegans (C. elegans)

#### 2.8.1. *C. elegans* Strains and Maintenance

The *C. elegans* strain was acquired from the Caenorhabditis Genetics Center (CGC, University of Minnesota, Minneapolis, MI, USA). The wild-type strain (Bristol, N2) was cultured on Nematode Growth Medium (NGM) agar plates supplemented with *Escherichia coli* (*E. coli*) OP50 as their food source [44]. The selection of worms at a precise developmental stage followed a synchronization process. Gravid hermaphrodites were subjected to a solution containing 1% NaOCl and 0.25 M NaOH to induce rupture of the worms’ cuticle, releasing the eggs and thereby ensuring synchronous development. Worms were selected for use at specific larval stages, as indicated in the tests described below. 

#### 2.8.2. Survival Assay 

The survival assay was conducted using worms at the young adult stage. Initially, worms in the L1 larval stage were plated with *E. coli* OP50 as a food source and incubated until they reached young adulthood at 20 °C for subsequent analysis. A total of 100 worms at the young adult stage were exposed to 4-PSCO free (300 μg/mL), NC P, and 4-PSCO NC at concentrations of 50, 150, and 300 μg/mL for 1 h. The worms were gently touched with a platinum wire, and animals that did not respond were classified as dead. All groups were evaluated in three independent experiments [45].

#### 2.8.3. Behavioral Tests

Behavioral tests, such as pharyngeal pumping, head thrash, defecation cycle, and touch response tests, were performed after 1 h of exposure of the animals to free 4-PSCO (300 μg/mL) or formulations NC P and 4-PSCO NC (300 μg/mL).

##### Pharyngeal Pumping and Head Thrashes 

After treatment, the animals were washed three times with M9 buffer and transferred to plates containing NGM and *E. coli* OP50 as a food source for a 1-h adaptation period. Subsequently, the worms were used for pharyngeal pumping and head thrash assays. In the pharyngeal pumping test, the number of pharyngeal contractions in each worm on the plate was counted for 10 s using a microscope [46]. For the head thrash test, worms were randomly selected and placed in a drop of M9 buffer for 1 min. Then, the number of head movements in each worm was counted for 20 s [47]. Five worms per group per experiment were assessed in three independent experiments (n = 15). The values for each worm were averaged and expressed as a percentage of control for both pharyngeal pumping and head thrashes.

##### Touch Response

In the tactile response test, worms were evaluated by delicately touching the head region using a bristle brush. A backward movement indicated a positive response, whereas animals that did not react were considered to have no adverse reactions to the stimulation. Each worm underwent a total of 10 touches, with a 10-s interval between each trial. Four assays were carried out at different intervals, with 5 worms analyzed in each experiment [48].

##### Defecation Cycle

The defecation cycle assay was evaluated by observing the interval time between intestinal contractions. The defecation behavior of young adult worms involves a sequence of anterior and posterior contractions followed by an expulsion event. The average of three intervals was measured in each animal in a total of 15 animals per group [49].

### 2.9. Evaluation of Short-Term Oral Toxicity in Mice

Prior studies suggested a gender-related difference in the ability to respond to oxidative stress, one of the parameters that may lead to drug toxicity [50,51]. For this reason, we chose to study both genders. For evaluation of oral toxicity, male and female mice were randomly distributed into four groups (n = 5 animals/group) and received one daily administration, for seven days, of the respective treatments as follows: (I) Control: canola oil (10 mL/kg, i.g.); (II) NC P: NC P suspension (10 mL/kg, i.g.); (III) 4-PSCO free: 4-PSCO at a dose of 10 mg/kg dissolved in canola oil (10 mL/kg, i.g.); (IV) 4-PSCO NC: 4-PSCO at a dose of 10 mg/kg incorporated into the NC suspension (10 mL/kg, i.g.). 

The 4-PSCO was dissolved in canola oil (10 mL/kg), and the formulations (4-PSCO NC and NC P) were prepared according to the previously described procedures and used without any modification. The animals received the treatment via the i.g. route in a repeated administration regimen with a constant volume relative to their body weight (10 mL/kg). Each administration corresponded to a 4-PSCO dose of 10 mg/kg regardless of its form. We selected this dose based on previous research that demonstrated the pharmacological effects of organoselenium compounds and coumarin derivatives, encompassing antinociceptive and anti-inflammatory properties [4,52]. The treatment regimen was designed based on previous studies that evaluated the short-term toxicity of the compounds of interest, both in their free and encapsulated forms, as well as the alterations caused by the repeated administration of these compounds [23,53,54].

The animals underwent regular monitoring at 24 h intervals to record occurrences of mortality, changes in behavioral patterns, ptosis, tremors, diarrhea, salivation, piloerection, locomotor alterations, or seizures. Additionally, before initiating treatment, we recorded their body weight and food consumption. Daily monitoring of these parameters continued during the experimental protocol (Figure 3).

#### 2.9.1. Toxicological Parameters 

On the 8th day of the experimental protocol, 24 h after the last treatment, the animals were slightly anesthetized with isoflurane before blood collection through heart puncture. The collected samples, treated with heparin, were then centrifuged at 2500 rpm for 10 min to obtain the plasma fraction, which was utilized to determine hepatic and renal function markers using commercial kits. Following this procedure, mice were euthanized with an overdose of isoflurane anesthetic, and some tissues (including the brain, spinal cord, spleen, liver, and kidneys) were extracted for macroscopic evaluation. The brain and spinal cord were immediately frozen and stored at −80 °C after extraction. The spleen, liver, and kidney tissues were quickly weighed to estimate the organ/body weight ratio and for physical change analysis. Subsequently, they were frozen and stored at −80 °C for biochemical assays. The organ/body weight ratio for the spleen, liver, and kidneys was calculated as (organ weight (g)/body weight of the animal on the day of sacrifice (g)) × 100 [55].

##### Tissue Processing for Biochemical Analyses

Brain, spinal cord, liver, and kidneys were processed to determine thiobarbituric acid reactive species (TBARS) and non-protein thiol (NPSH) levels. The tissue samples were homogenized in 50 mmol L-1 Tris HCl, pH 7.4 (using a ratio of 1 part tissue to 10 parts volume), and then centrifuged at 3000 rpm for 10 min to obtain a supernatant (S_1_). Freshly prepared S_1_ was used for the analysis [56,57]. 

TBARS levels

TBARS levels were assessed following the method outlined by Ohkawa et al. [58], which is commonly employed for measuring lipid peroxidation. In this procedure, 200 µL of S_1_ was incubated with 0.8% thiobarbituric acid, acetic acid buffer (pH 3.4), and 8.1% sodium dodecyl sulfate (SDS) at 95 °C for 2 h. The resulting color reaction was measured at 532 nm, and TBARS levels were quantified and expressed as nmol MDA/mg protein.

Non-protein thiol content

Ellman’s method [59] was employed to determine the NPSH content, a non-enzymatic antioxidant defense. Initially, a sample of S_1_ was combined with 10% trichloroacetic acid (TCA) in a 1:1 ratio. Following centrifugation at 3000 rpm for 10 min, the protein pellet was discarded, and the clear supernatant was utilized to assess free thiol groups. A portion of the supernatant (200 μL) was then mixed with 1 M potassium phosphate buffer (pH 7.4) and 10 mM 5,5′-dithiobis-2-nitrobenzoic acid (DTNB). The resultant color reaction was measured at 412 nm, and NPSH levels were quantified and expressed as nmol of NPSH/g tissue.

Plasma hepatic and renal biochemical markers

The activities of alanine aminotransferase (ALT) and aspartate aminotransferase (AST), as well as the levels of urea, were determined using a kinetic and endpoint colorimetric method, respectively (BIOCLIN; LABTEST, Lagoa Santa, Brazil).

### 2.10. In Vivo Studies 

#### 2.10.1. Behavioral Tests

##### Glutamate-Induced Nociception

Mice were pre-treated with different substances depending on the groups that they belonged to: canola oil (control group, 10 mL/kg, i.g.), 4-PSCO free (0.1, 1, and 5 mg/kg, i.g.), meloxicam (5 mg/kg, i.g.), NC P (10 mL/kg, i.g.), or 4-PSCO NC (1 mg/kg, i.g.). The dose of meloxicam (5 mg/kg) was selected to compare with the highest dose tested for the 4-PSCO compound. The glutamate (20 mmol/paw, 20 μL, intraplantar (i.pl.)) was injected into the right hind paw, while 0.9% saline (20 μL, i.pl.) was injected into the left hind paw (control) 30 min after pre-treatment. The animals were placed in individual boxes and observed for 15 min after the injection of the phlogistic agent. The duration of paw licking in response to the glutamate injection was considered an indicator of nociceptive behavior [60]. After the observation period, the animals were euthanized, and the paws were collected for the determination of edema, which was evaluated by comparing the weight difference between the glutamate-injected paw and the contralateral paw weight (treated with 0.9% saline). The results were represented as licking (s) for the glutamate test and weight (mg) for paw edema.

##### Time–Response Curve of 4-PSCO and 4-PSCO NC on Mechanical Withdrawal Threshold Induced by Complete Freund’s Adjuvant

To evaluate the time–response curve, the animals received an i.pl. injection of CFA (1 mg/mL Mycobacterium tuberculosis, 20 μL) in the right hind paw and saline solution (0.9%, 20 μL/paw) in the left paw. After 24 h, animals received canola oil (control group, 10 mL/kg, i.g.), 4-PSCO free (1 mg/kg, i.g.), NC P (10 mL/kg, i.g.), or 4-PSCO NC (1 mg/kg, i.g.). The nociceptive response was verified using the digital analgesimeter at 30 min and 2, 4, 6, and 24 h after treatment. This test was performed to measure the animals’ mechanical hyperalgesia through the paw withdrawal threshold. To conduct the test, the mice were individually placed in acrylic cages with wire grid floors in a quiet and acclimatized room. The test involved evoking a hind paw flexion reflex using an anaesthesiometer (Insight, Ribeirão Preto, SP, Brazil) equipped with a polypropylene tip following the method described by Alamri et al. [61] with some modifications. Before CFA induction, the animals underwent a baseline evaluation of mechanical sensitivity to ensure accurate results during the behavioral test. The data were expressed as the withdrawal threshold (g).

##### Hot Plate Test

For this test, mice were placed on a heated metal plate maintained at 55 ± 1 °C [62]. The latency of nociceptive responses, such as licking or shaking one of the paws or jumping, was measured. The maximum cutoff time was set at 45 s, and thermal withdrawal latencies were measured before treatment and 30 min after treatment with the following substance: canola oil (control group, 10 mL/kg, i.g.), 4-PSCO free (0.1, 1, and 5 mg/kg, i.g.), morphine (5 mg/kg, i.p.), NC P (10 mL/kg, i.g.), or 4-PSCO NC (1 mg/kg, i.g.). The delta latency (Δt) was calculated for each animal: Δt (s) = post-treatment latency − pre-treatment latency.

##### Open-Field Test

The OFT was utilized to assess the locomotor and exploratory behaviors of mice. Constructed from plywood, the open field featured walls measuring 30 cm in height, with the floor divided into nine squares using masking tape markers (3 rows of 3). During the test, each animal was positioned at the center of the open field and observed for 4 min to record locomotor activity (number of segments crossed with the four paws) and exploratory behavior (number of rearings on the hind limbs). Following each session, the arena was sanitized with 20–30% ethanol, and individual mice were subjected to testing only once [63].

### 2.11. Statistical Analysis

The normality of data was evaluated by the D’Agostino and Pearson omnibus normality test. Statistical analysis was performed using GraphPad Prism 5.0. Data were analyzed by unpaired Student’s *t*-test, one-way ANOVA followed by Tukey’s test, or one-way ANOVA followed by Dunnett’s post hoc test when appropriated. Student’s *t*-test and one-way ANOVA followed by Tukey’s test were used to evaluate physicochemical parameters of the formulations and behavioral tests in mice. The one-way ANOVA followed by Dunnett’s post hoc test was used to evaluate toxicity parameters in *C. elegans*. Data from the experiments were expressed as mean ± standard error medium (S.E.M.). Values of *p* < 0.05 were considered statistically significant.

## 3. Results

### 3.1. Molecular Docking Analyses

Docking simulations were performed to analyze potential interactions between the 4-PSCO and protein targets, including MAPK, NFkB, PAD4, PI3K, JAK2, and TLR4. The binding free energy (Δ_G_) values resulting from the ligand–target interactions are summarized in Table 1. Negative values indicate favorable ligand–protein interactions.

The values presented in Table 1 demonstrate an affinity between 4-PSCO and the following protein targets: p38 mitogen-activated protein kinase (MAPK), peptidyl arginine deiminase type 4 (PAD4), phosphoinositide 3-kinase γ isoform (PI3K), Janus kinase 2 (JAK2), toll-like receptor 4 (TLR4), and nuclear factor-kappa β (NFkB).

Contact maps were generated to characterize the interactions between the ligand and targets, as depicted in Figure 4. The 4-PSCO predominantly interacts with JAK through the JAK PTK domain, where the ligand engages primarily in hydrophobic interactions facilitated by its aromatic moiety, as observed in other systems [28]. The interactions between 4-PSCO and JAK mainly involve π–sigma interactions with residues Leu983, Val863, and Leu855 and π–alkyl interactions with Ala880, Leu932, and Met929 (Figure 4A).

Similar hydrophobic interactions were observed with TLR4. The 4-PSCO occupies the MD-2 binding pocket, akin to analogous systems [64]. However, due to its smaller size compared to other ligands, it tends to interact predominantly with the hydrophobic region of the protein as it cannot reach the polar region. Notable interactions with TLR4 include π–sigma interactions with residues Ile52 and Val48 and π–alkyl interactions with residues Leu61, Ile32, and Ile153. Additionally, other π interactions were observed with TLR4, such as π–π stacked interactions with Phe119 and Phe151 and pi–sulfur interactions with Cys133 (Figure 4B).

Hydrogen bond interactions between the oxygen of coumarin and the targets NFkB and PI3K were observed. For NFkB, the ligand is positioned in the polar region responsible for the interaction of NFkB with DNA, forming a hydrogen bond with residue Leu207, which is consistent with a similar compound [65]. However, due to the limited presence of hydrogen donor or acceptor groups in 4-PSCO, its interaction is less favorable. Hydrophobic interactions are the predominant interactions in these cases, with compound 4-PSCO engaging with residues Lys241, Asp239, Ser208, Tyr57, Thr143, Thr153, His141, and Lys144 in NFkB (Figure 4C).

In PI3K, a hydrogen bond was observed with residue Tyr867. However, hydrophobic interactions also play a significant role in these cases. Other π interactions were observed in PI3K with residues Ala885, Met804, Ile831, Ile881, Met953, and Trp812, for example. In this case, hydrophobic interactions are primarily responsible for the ligands’ stability in the protein’s binding site, consistent with other systems [32] (Figure 4D).

The ligand interacts with MAPK through the ATP-binding site, situated between hydrophobic regions I and II, but differently compared to other inhibitors [27]. The small size and lack of planarity of the molecule as a whole hinder its hydrogen interactions with residues Gly110 and Met109. Instead, the 4-PSCO forms a hydrogen bond with residue Asp112 and exhibits van der Waals interactions with residues Lys53, Leu167, Met109, and Tyr35, for example (Figure 4E). The results of interaction maps between the reference drugs (ketoprofen, naproxen, tofacitinib, dexamethasone, methotrexate) and the proteins MAPK, PAD4, PI3K, JAK2, TLR4, and NFκB are presented in the Appendix A.

The 4-PSCO interacts with the C-terminal domain binding site of PAD4 [31], engaging in hydrophobic interactions with Arg374, Arg372, and Val469, among others. However, as the binding of substrates in the enzyme primarily occurs through hydrogen bond interactions [31] and flexible substrates, the molecule exhibits a weak fit within the PAD4 pocket site (Figure 4F).

Overall, π interactions and hydrogen bonds are the main contributors to the affinity values observed for 4-PSCO. The binding of 4-PSCO to targets that require flexible hydrophobic interactions and fewer hydrogen bonds may be favorable. The frontier orbitals HOMO and LUMO for 4-PSCO (Figure 5) indicate that the perpendicular orientation of the chromenone and organoselanyl moieties limits extensive conjugation across the entire molecule.

### 3.2. Nanocapsule Suspension Physicochemical Characterization and Stability Evaluation

The results of the physicochemical characterization and stability evaluation are shown in Table 2. The formulations presented a milky appearance and were macroscopically homogeneous with no visible precipitation. The suspensions of 4-PSCO NC and NC P exhibited an average particle size of approximately 160 nm, displaying a narrow size distribution, as indicated by a PDI value < 0.120. Volume-weighted mean diameters (D[4;3]) determined by laser diffraction were lower than 0.160 µm (0.158 ± 0.002), accompanied by SPAN values lower than 2 (0.678 ± 0.048). Analysis of the granulometric profile revealed unimodal size distribution curves, with diameters predominantly in the nanometric range at their initial time. Appendix A contains representative images of the granulometric profile for both the 4-PSCO NC and NC P. Moreover, all formulations exhibited a negative zeta potential and maintained a slightly acidic pH. The 4-PSCO content and encapsulation efficiency reached 100%. The unpaired Student’s *t*-test, employed for statistical analysis, demonstrated no significant differences among the formulations concerning all tested parameters (*p* > 0.05).

Following a 30-day storage period at room temperature, the formulations exhibited an unchanged macroscopic appearance, showing no visible alterations. Furthermore, no statistically significant differences were observed in any of the physicochemical parameters tested (size, PDI, ZP, pH, 4-PSCO content) compared to the initial measurements (*p* > 0.05; one-way ANOVA of repeated measures).

Considering in vitro release profiles, the analysis showed that nanocapsules promoted 4-PSCO controlled release starting at 1 h, as shown in Figure 6. In 8 h, 4-PSCO was completely released from a methanolic solution, whereas only 20% of 4-PSCO was released from the nanocapsules. This demonstrates the nanocarrier’s ability to extend drug release, which was corroborated by the fit of the experimental data to the zero-order equation (r > 0.99).

### 3.3. Effects of 4-PSCO Free and 4-PSCO NC in Toxicity Assay in C. elegans

The effects of 4-PSCO free, NC P, and 4-PSCO NC in the survival assay on *C. elegans* are shown in Figure 7. Exposure to NC P or 4-PSCO NC (at concentrations of 50, 150, and 300 μg/mL) for 1 h did not affect the survival of *C. elegans* (*p* > 0.05). However, when nematodes were exposed to a concentration of 300 μg/mL of the 4-PSCO free, there was a decrease in the number of live worms compared with the control group (ANOVA: F _(5, 12)_ = 2.434, *p* < 0.05) (Figure 7).

The results presented in Figure 8A–D pertain to the behavioral tests conducted on *C. elegans*. Exposure of young adult worms to 4-PSCO free (300 μg/mL), 4-PSCO NC (300 μg/mL), and NC P did not induce changes in the motility parameters assessed in the pharyngeal pumping test (ANOVA: F _(3, 56)_ = 2.574, *p* > 0.05) (Figure 8A) and the head thrashes (ANOVA: F _(3, 56)_ = 1.530, *p* > 0.05) (Figure 8B) of the animals (*p* > 0.05). However, in Figure 8C, a significant decrease in touch response can be observed in the Bristol N2 strain following treatment with 4-PSCO NC at 300 μg/mL compared with the control group (ANOVA: F _(3, 76)_ = 2.087, *p* < 0.05) (Figure 8C). No significant differences were observed in touch response after treatment with 4-PSCO free (300 μg/mL) and NC P (Figure 8C). In the defecation cycle test (ANOVA: F _(3, 56)_ = 2.650, *p* > 0.05) (Figure 8D), no significant difference was observed among young adult worms treated with 4-PSCO free (300 μg/mL), NC P, or 4-PSCO NC (300 μg/mL) (*p* > 0.05) (Figure 8D).

### 3.4. Repeated Administration of 4-PSCO (Free or Encapsulated) Did Not Induce Toxicity in Mice

Repeated treatment with 4-PSCO free and 4-PSCO NC did not induce alterations in the clinical signs or mortality of the animals. No change in behavioral patterns, ptosis, tremors, diarrhea, salivation, piloerection, or seizures was noted. Additionally, statistical analysis did not reveal significant differences in food/water consumption and body weight between male and female mice in the evaluated groups (Appendix A).

Following 7 days of treatment with the samples, no significant change was observed in the absolute and relative organ weights (spleen, kidneys, liver) or macroscopic characteristics among the experimental groups in male and female mice (Appendix A). The results of the statistical analyses of food/water consumption, body weight, and absolute organ weights are presented in the Appendix A. 

#### 3.4.1. Effect of Repeated Treatment with 4-PSCO Free and 4-PSCO NC on Lipid Peroxidation and Non-Protein Thiol Levels 

The effects of 4-PSCO free and/or 4-PSCO NC on TBARS levels in the brain, spinal cord, liver, and kidneys of female and male mice are presented in Appendix A. The results indicate that repeated treatment with 4-PSCO free (10 mg/kg) and 4-PSCO NC (10 mg/kg) did not alter the TBARS levels in the evaluated tissues in both sexes in comparison to those of the control and NC P groups, respectively. Similarly, continuous administration of NC P (10 mL/kg) did not alter this parameter in the brain, liver, and kidneys of male and female mice (Appendix A). 

Our findings regarding NPSH levels are demonstrated in Appendix A. Repeated treatment with 4-PSCO free (10 mg/kg) and 4-PSCO NC (10 mg/kg) did not alter NPSH levels in the brain, spinal cord, liver, and kidneys of male and female mice compared to those of control and NC P groups. Similarly, the NC P (10 mL/kg) treatment also had no effect on this parameter in the evaluated tissues in both sexes. Furthermore, the NPSH levels in the tissues were not different between 4-PSCO free and 4-PSCO NC groups (Appendix A). The results of the statistical analyses of TBARS and NPSH levels are presented in the Appendix A.

#### 3.4.2. Repeated Treatment with 4-PSCO Free and 4-PSCO NC Did Not Alter Plasma Biochemical Parameters of Renal and Hepatic Function

The results of the biochemical parameters ALT, AST, and urea analyzed in the plasma of male and female mice are showed in the Appendix A. The repeated treatment with 4-PSCO free and/or 4-PSCO NC did not alter the activity of the AST and ALT and the levels of urea in the plasma of male and female mice. The results of the statistical analyses are presented in the Appendix A. 

### 3.5. The 4-PSCO Free and 4-PSCO NC Significantly Reduce Nociception and Paw Edema Induced by Glutamate

Figure 9 illustrates the effects of the administration of 4-PSCO free and 4-PSCO NC on behavioral response in the glutamate test in male and female mice. Data analysis revealed that 4-PSCO free and 4-PSCO NC significantly decreased the nociceptive response compared to that of the control and NC P groups in male mice, respectively (Figure 9A). The 4-PSCO free, at doses of 1 and 5 mg/kg, reduced glutamate-induced licking time by 63% and 78%, respectively. The 4-PSCO NC (1 mg/kg) reduced the licking time by 29% (F _(6, 49)_ = 26.42, *p* < 0.0001) compared with that of the NC P group. However, 4-PSCO free (0.1 mg/kg) and meloxicam (5 mg/kg) did not reduce the glutamate-induced nociceptive response compared with that of the control group (Figure 9A).

Similarly, in female mice, 4-PSCO free and 4-PSCO NC caused a significant decrease in the nociceptive response compared with that of the control and NC P groups, respectively. At doses of 1 and 5 mg/kg, 4-PSCO free effectively reduced glutamate-induced licking time by 50% and 85%, respectively. The 4-PSCO NC (1 mg/kg) reduced the licking time by 27% (F _(6, 49)_ = 22.23, *p* < 0.0001) compared with that of the NC P group. Again, 4-PSCO free (0.1 mg/kg) and meloxicam (5 mg/kg) did not reduce the glutamate-induced nociceptive response (Figure 9B). Additionally, it was observed that 4-PSCO free had a stronger antinociceptive effect compared to 4-PSCO NC (Figure 9B).

Furthermore, 4-PSCO free, at the doses of 1 and 5 mg/kg, significantly reduced glutamate-induced paw edema formation in male mice by 18 and 19% compared with that of the control group, respectively (male mice). The 4-PSCO NC (1 mg/kg) also resulted in a significant reduction (23%) in glutamate-induced paw edema formation in male mice compared to that of the NC P group (F _(6, 49)_ = 10.07, *p* < 0.0001) (Figure 9C). It is noteworthy that the formation of paw edema in male mice exhibited a significant difference between the treatment with 4-PSCO NC (1 mg/kg) and the free compound at all studied doses (4-PSCO free, 0.1, 1, and 5 mg/kg) (Figure 9C). In female mice, the administration of 4-PSCO free resulted in a reduction in paw edema formation of 32% (1 mg/kg) and 33% (5 mg/kg) compared with that of the control group (F _(6, 49)_ = 3.829, *p* < 0.001) (Figure 9D). However, 4-PSCO free, at a dose of 0.1 mg/kg, and meloxicam (5 mg/kg) were unable to reduce glutamate-induced edema in both sexes. Similarly, treatment with 4-PSCO NC in female mice also did not exhibit an anti-edematogenic effect (*p* > 0.05) (Figure 9D). In male and female mice, no statistical difference was observed in the licking time and paw edema between the two control groups, the vehicle, and the NC P.

### 3.6. The 4-PSCO NC Is More Effective than the Free Compound in Reducing CFA-Induced Mechanical Hypersensitivity

The results depicted in Figure 10 demonstrate the antinociceptive effect exerted by 4-PSCO free and 4-PSCO NC against the inflammatory process triggered by CFA in the mechanical sensitivity test. Male mice treated with 4-PSCO free (1 mg/kg) exhibited a 65% reduction in mechanical hypersensitivity induced by CFA compared to that of the control group (Figure 10A). For the 4-PSCO NC, a maximum inhibition of 35% was observed in mechanical hypersensitivity induced by CFA compared to that of the NC P group after 0.5 h of treatment (Figure 10A). From 0.5 h to 72 h after treatment, both 4-PSCO free and 4-PSCO NC continued to exhibit an antinociceptive effect compared to that of the CFA-treated group (Figure 10A). Similarly, in female mice, treatment with 4-PSCO free and 4-PSCO NC reduced the mechanical sensitivity induced by CFA starting at 0.5 h, and this effect persisted for up to 72 h (Figure 10B). In 72 h, 4-PSCO free showed a maximum inhibition of 46% and 4-PSCO NC a maximum inhibition of 51% compared with that of the control and NC P groups, respectively. It is noteworthy that, at 48 h and 72 h, 4-PSCO NC was more effective in reducing hypersensitivity to mechanical stimuli induced by CFA than the free-form compound in male mice (ANOVA: F _(21, 224)_ = 20.84, *p* < 0.0001) (Figure 10A). In female mice, this difference was only observed in 72 h, and 4-PSCO NC showed a maximum inhibition of 50% compared with that of the NC P group (ANOVA: F _(21, 224)_ = 42.40, *p* < 0.0001) (Figure 10B).

### 3.7. Acute Treatment with 4-PSCO Free and 4-PSCO NC Increases Thermal Stimulus Latency

The effects of acute treatment with 4-PSCO free and 4-PSCO NC in response to latency to thermal stimulus in the hot plate test in male and female mice are demonstrated in Figure 11A,B. Acute treatment with 4-PSCO free (1 and 5 mg/kg) in male mice increased the response latency to the thermal stimulus by 61% and 67% compared with that of the control group, respectively (Figure 11A). The 4-PSCO NC (1 mg/kg) increased the response latency to thermal stimulus by 97% compared with that of the NC P male mice (F _(6, 49)_ = 12.01, *p* < 0.0001) (Figure 11A).

In female mice, both 4-PSCO free and 4-PSCO NC exhibited significant antinociceptive activity. At doses of 1 and 5 mg/kg, 4-PSCO free increased the latency time to the thermal stimulus response compared with that of the control group. The 4-PSCO NC increased the latency time to the thermal stimulus response by 82% compared to that of the NC P group (F _(6, 49)_ = 15.66, *p* < 0.0001) (Figure 11B). Morphine, a drug with a central effect used as a reference in the hot plate test, at a dose of 5 mg/kg, effectively induced an antinociceptive response to thermal stimulation compared with that of the control group in both sexes. However, 4-PSCO free (0.1 mg/kg) did not elicit a greater antinociceptive response to thermal stimulation compared to that of the control group in male and female mice. The morphine (5 mg/kg) was more effective in increasing the nociceptive response to thermal stimulus than 4-PSCO free (0.1 mg/kg) and 4-PSCO NC (1 mg/kg) in female mice (Figure 11A,B).

### 3.8. The Locomotor and Exploratory Domains Did Not Change with 4-PSCO Free and 4-PSCO NC Treatment

The possible effects of treatments on locomotor and exploratory activities of male and female mice were evaluated in the OFT (Appendix A). The data analysis of OFT revealed that no alteration in the number of crossings (ANOVA: F _(7, 56)_ = 0.9080, *p* > 0.05) and rearing (ANOVA: F _(7, 56)_ = 1.015, *p* > 0.05) was observed after the different treatments in male mice (Appendix A). Similarly, the treatments with 4-PSCO free and 4-PSCO NC did not cause any significant change in the number of crossings (ANOVA: F _(7, 56)_ = 1.456, *p* > 0.05) or rearing (ANOVA: F _(7, 56)_ = 1.014, *p* > 0.05) in female mice (Appendix A).

## 4. Discussion

This study marks the first exploration of the promising interactions between the 4-PSCO molecule—an organic selenium compound incorporating coumarin—and proteins integral to intracellular signaling pathways associated with pain and inflammation, as demonstrated through computational tests. Additionally, compelling evidence is presented here for the first time, indicating that the oral administration of free and nanoencapsulated 4-PSCO at a low dose elicits significant antinociceptive and anti-inflammatory responses in acute nociception models. It is worth noting that the effects of the treatment persist for up to 72 h. In particular, 4-PSCO NC is more effective than free 4-PSCO in treating the pain caused by CFA over time. Additionally, the study confirms the stability of the developed formulations and the absence of any toxic effects associated with 4-PSCO NC. 

Preclinical studies have explored the effects of organoselenium compounds, particularly their antioxidant [66], anti-inflammatory, and antinociceptive properties [67,68,69,70]. However, our research adds value by elucidating the specific advantages and contributions of our approach to the seleno–coumarin compound. The advantage lies in the incorporation of the coumarin nucleus into the molecule. Numerous studies support and demonstrate the advantages and biological effects of the coumarin derivates [71,72]. Although organoselenium compounds have demonstrated efficacy, the inclusion of the coumarin nucleus further substantiates the observed effects. Planar aromatic rings linked with lactone functionality and hydrogen bonds play a crucial role in protein–ligand interactions, enhancing cell recognition and desired pharmacological effects [73]. Structural modifications at positions C-3, C-4, and C-7 further amplify the biological activities of synthetic coumarins [74].

Docking simulations have unveiled significant interactions between the 4-PSCO molecule and target proteins, namely MAPK, JAK2 (Δ_G_ of −7.9 kcal/mol), TLR4 (Δ_G_ of −7.8 kcal/mol), NFkB (Δ_G_ of −5.9 kcal/mol), PAD4 (Δ_G_ of −6.7 kcal/mol), and PI3K (Δ_G_ of −7.7 kcal/mol), with MAPK exhibiting the highest affinity (Δ_G_ of −8.3 kcal/mol). Negative values signify advantageous interactions between the ligand and the protein. The proposed interaction mechanism between 4-PSCO and the analyzed proteins involves an antagonistic mode, wherein 4-PSCO anchors to specific binding sites on each receptor [29,75,76]. Notably, the 4-PSCO molecule has demonstrated promising interactions with essential amino acids critical for the function of these proteins, including Leu932, Met109, Lys53, and Tyr867 [77,78]. These amino acids are pivotal in intracellular signaling pathways associated with pain, inflammation, cell differentiation, proliferation, and survival [76,79]. 

Indeed, it is crucial to note that our investigations revealed that 4-PSCO interactions with the target proteins are comparable or superior to those of the reference drugs used in pain and inflammation conditions. The only exception was the interaction with PAD4, which yielded a result inferior to that observed with tofacitinib. These findings align with the existing literature, where recent studies have explored the affinity of drugs belonging to the classes of disease-modifying, anti-rheumatic, and nonsteroidal anti-inflammatory drugs with the same proteins examined in our study [80,81,82,83]. The consistency in results suggests that 4-PSCO shares similar binding affinities with the reference drugs. Thus, these findings suggest that 4-PSCO is able to interact and negatively modulate these proteins, reinforcing the potential for improved clinical outcomes in patients with inflammatory diseases.

Enhancing drug delivery systems using nanosystems presents effective and safe treatment options. The development of polymeric nanocapsules in this study employed the interfacial deposition method, a simple, rapid, and cost-effective approach applicable to a wide range of drugs and molecules [22]. The ethylcellulose polymer chosen for this investigation possesses pertinent properties such as particle formation, flexibility, and mechanical resistance, and is particularly suitable for encapsulating lipophilic substances [84,85]. The lipophilic nature of the 4-PSCO molecule is recognized for its association with critical criteria such as gastrointestinal permeability, blood–brain barrier passage, and plasma protein binding [86]. Indeed, it has already been described in the literature that the presence of the selenium atom, as well as the coumarin nucleus in the structure of organic compounds, gives them high lipophilicity [87,88]. In this study, the lipophilic properties of 4-PSCO, coupled with the selection of ethylcellulose polymer, aligned with the promising effects of the formulated nanocapsules. Notably, the encapsulation efficiency of 4-PSCO reached an outstanding 100%, signifying a highly promising and satisfactory outcome. The macroscopic evaluation revealed homogeneity without visible precipitation or micrometric population, indicating a unimodal particle distribution profile [89]. As demonstrated in the present formulations, polymeric nanoparticles of subcellular size can modulate the pharmacokinetic properties of various active substances [22,90]. Moreover, they protect the drug during transportation to the target tissue, thereby reducing the required therapeutic concentration and minimizing drug-induced side effects and toxicity [22,90,91].

From this perspective, despite the notable pharmacological effects of organoselenium compounds and coumarin derivatives, their potential is often overshadowed by toxicity issues. Evidently, in the survival assay conducted in *C. elegans*, exposure to the free compound reduced the number of live worms, a reliable indicator of toxic effects [92]. Aligning with our objective of enhancing compound safety through nanotechnology, the treatment with 4-PSCO NC did not exhibit any noticeable alterations in the survival assay in *C. elegans*. These findings suggest that encapsulating chemical molecules in nanostructures holds promise in mitigating toxic effects, as indicated by previous studies [23,93]. Conversely, when worms were exposed to a higher concentration of 4-PSCO NC, their touch response was reduced compared to that of the control group. The touch receptor neuron, responsible for sensing mechanical stimuli, plays a crucial role in modulating locomotion, feeding behaviors, egg posture, and excretion [94]. Interestingly, while the touch behavior was specifically affected in worms, other parameters remained unaffected following exposure to encapsulated 4-PSCO. 

Considering these results, further studies are warranted to understand the effects of 4-PSCO NC on locomotor capacity comprehensively. While the *C. elegans* model is a valuable tool for estimating toxicological effects, it has limitations [75]. Therefore, in vivo studies are imperative to investigate new formulations’ potential toxicity and metabolism [95]. This study conducted the open-field test to expand our understanding of the effects of 4-PSCO NC on locomotor and exploratory activities. The results demonstrated that neither pharmaceutical form altered the number of crossings and rearings in the open-field test in male and female mice, indicating no changes in the assessed capabilities. Throughout this study, the repeated administration posology of both 4-PSCO free and 4-PSCO NC showed no signs of toxicity or mortality in the animals. Furthermore, plasma biochemical parameters and food and water intake remained unaffected, suggesting the absence of renal or hepatic alterations. While multiple parameters contribute to toxicity, oxidative stress is a significant mechanism supported by the literature [50,51]. In this study, oxidative stress evaluations, conducted through the TBARS test and NPSH levels, did not indicate tissue damage following repeated administration of both forms of 4-PSCO across different tissues and genders. These findings align with previous research, underscoring the antioxidant effects of organic selenium compounds [96].

In light of the results obtained from the toxicological assessment and recognizing the existing limitations in current treatments for pain and inflammation, we proceeded with our research endeavors to delve into the antinociceptive and anti-inflammatory effects of 4-PSCO, aiming to enhance our understanding of its therapeutic potential. The nociceptive response induced by i.pl. glutamate injection holds significance in acute and chronic pain processing, activating neurons from the dorsal horn to the spinal cord and contributing to releasing cytokines and chemicals associated with inflammation [97]. In our study, administering low doses of both 4-PSCO free and 4-PSCO NC remarkably diminished glutamate-induced licking and edema formation. Notably, 4-PSCO NC exhibited superior efficacy in reducing paw inflammatory alterations compared to its free form in male mice. This observation is significant, as our current data underscore that the attenuation of the inflammatory process by 4-PSCO is crucial in preventing the amplification of this cascade, thereby highlighting its potent and effective antinociceptive and/or anti-inflammatory effects. These findings are supported by a previous study that demonstrated that the encapsulation of organic selenium compounds enhances their antinociceptive action in preclinical models [98].

Within the scope of this investigation, our findings underscore the rapid onset of action in the mechanical hypersensitivity test induced by i.pl. injection of CFA, irrespective of the 4-PSCO form, with the effect enduring for up to 72 h. Notably, treatment with 4-PSCO NC exhibited a more pronounced antihypernociceptive effect, surpassing free 4-PSCO. The intense inflammatory pain triggered by CFA initiates tissue damage and the release of inflammatory agents, culminating in allodynic and hypernociceptive responses and the sensitization of pain-processing neurons [3,99]. These results align seamlessly with prior research, highlighting the sustained impact of nanocarriers compared to unencapsulated drugs across various animal models of nociception and inflammation, thus reinforcing the advantages of our study [20,21,42,98,100].

Indeed, in the present study, the antinociceptive action of 4-PSCO NC exhibited a significant difference compared to the free compound. The differences between groups were noted at 48 and 72 h for male mice and at 72 h for female mice. Multiple factors could contribute to the extended effect observed with 4-PSCO NC. These factors encompass the sustained release profile, broader biodistribution of 4-PSCO resulting in a higher cellular concentration, favorable solubility within the formulation matrix, and the diminutive size of the suspended nanoparticles. However, the absence of a pharmacokinetic study of 4-PSCO, to better understand the compound’s bioavailability in the body, is one of the limitations of our study. Despite the absence of this study, all these aspects represent pivotal attributes deserving consideration and are consistent with results previously demonstrated by other researchers [21,101,102]. This prolongation of the antinociceptive effect contributes to mitigating the inflammatory process.

Our findings reveal that the free-form and encapsulated 4-PSCO increased the latency to thermal stimulus in the hot plate test compared to that of the control and NC P groups, respectively. Notably, morphine, a well-established potent analgesic, exhibited superior efficacy in inhibiting thermal nociception compared to both free 4-PSCO and 4-PSCO NC. While the antinociceptive effects of 4-PSCO in its free and encapsulated forms did not surpass those of morphine, the current results underscore a promising analgesic action of 4-PSCO, irrespective of its pharmaceutical form. Furthermore, the data presented align with prior studies indicating that organoselenium compounds and coumarin derivatives can modulate a diverse array of signaling pathways, particularly those associated with supraspinal responses. This suggests that free 4-PSCO and its encapsulated form potentially act as centrally acting analgesics [67,96,103].

## 5. Conclusions

These findings underscore the significance of functionalizing a coumarin compound with a selenium atom, an area of research that has been relatively underexplored, resulting in promising pharmacological properties. Molecular docking tests demonstrated that 4-PSCO has a high affinity for pain and inflammation-related receptors, suggesting its efficacy in modulating pain pathways. Moreover, it emphasizes the potential of a straightforward formulation containing 4-PSCO to effectively alleviate nociceptive and inflammatory pain while concurrently extending these therapeutic effects and mitigating undesirable effects, indicating its safety profile. Consequently, both the free compound and its encapsulated form emerge as pertinent therapeutic innovative approaches for managing inflammatory diseases and pain conditions, presenting effective treatments devoid of toxic effects. Building on our promising initial findings, future research aims to deepen our understanding of pain mechanisms and the pharmacological effects of free and nanoencapsulated 4-PSCO, elucidating the involvement of the inflammatory process and oxidative stress. Additionally, we emphasize our focus on exploring and highlighting the nanoencapsulated form of the compound, recognizing nanotechnology as a promising and rapidly growing field.

## Figures and Tables

**Figure 1 pharmaceutics-16-00269-f001:**
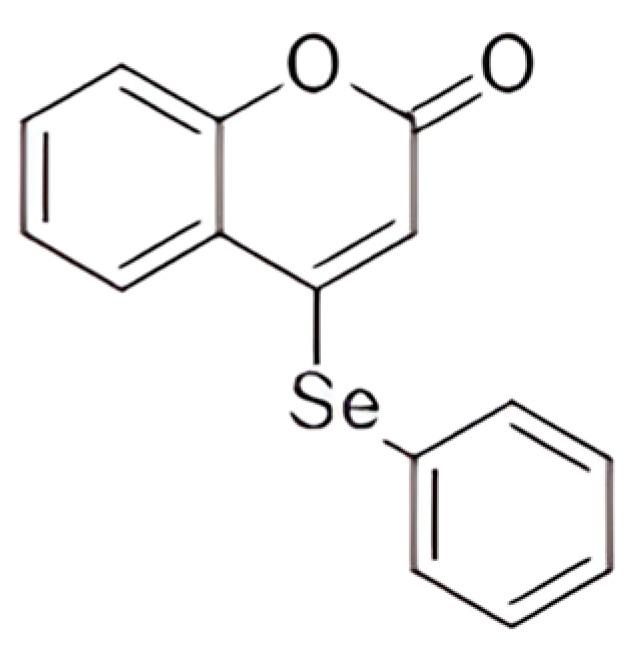
Chemical structure of 4-PSCO.

**Figure 2 pharmaceutics-16-00269-f002:**
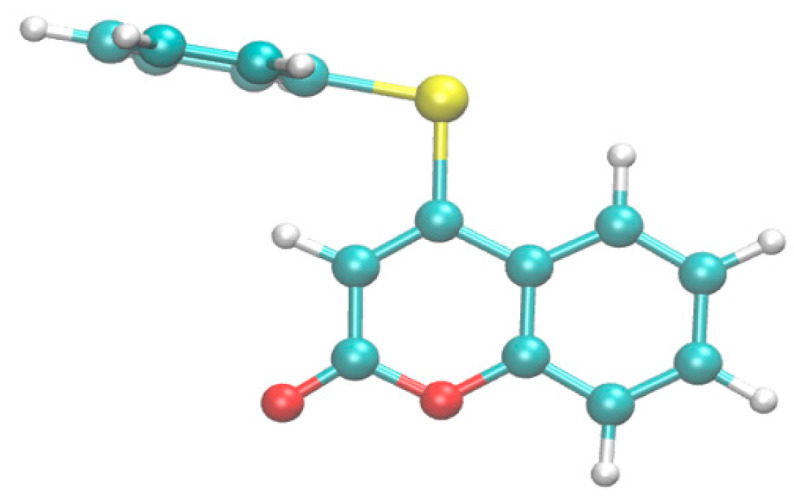
Initial conformation for compound 4-PSCO generated by CREST. The root-mean-squared deviation from a B3LYP/def2-TZVP optimized structure is 0.0154.

**Figure 3 pharmaceutics-16-00269-f003:**
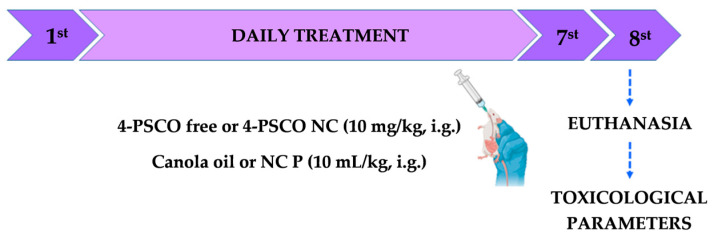
Experimental design of the toxicity protocol in mice.

**Figure 4 pharmaceutics-16-00269-f004:**
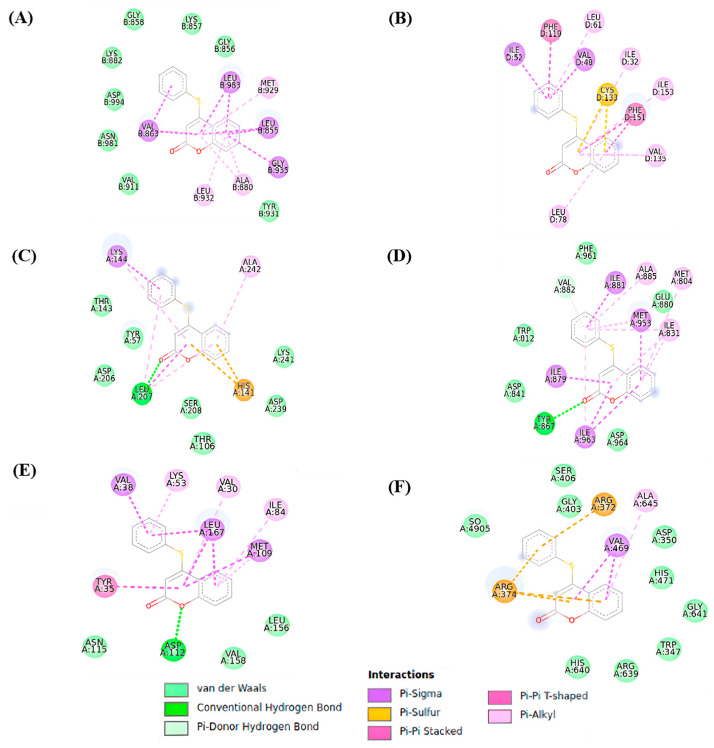
Selected contact maps of 4-PSCO with targets JAK (**A**), TLR4 (**B**), NFkB (**C**), PI3K (**D**), MAPK (**E**), and PAD4 (**F**).

**Figure 5 pharmaceutics-16-00269-f005:**
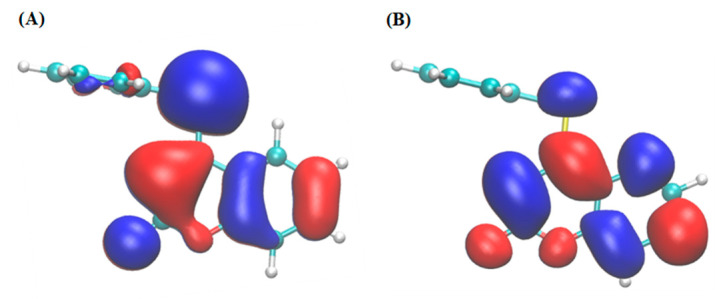
Frontier molecular orbitals HOMO (**A**) and LUMO (**B**) for 4-PSCO. Both orbitals are mainly localized in the chromenone moiety.

**Figure 6 pharmaceutics-16-00269-f006:**
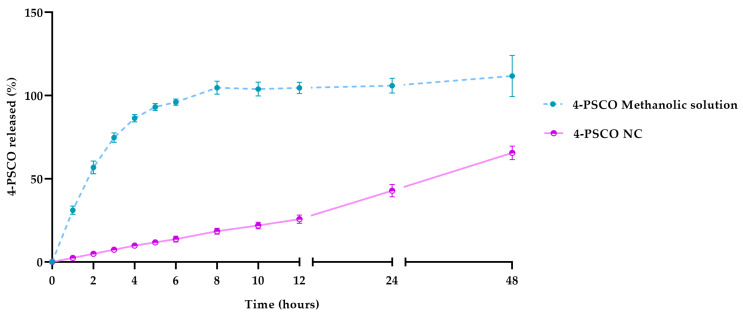
In vitro release profiles of 4-PSCO in free (4-PSCO methanolic solution) or nanoencapsulated form (4-PSCO NC). The values are reported as mean ± SD.

**Figure 7 pharmaceutics-16-00269-f007:**
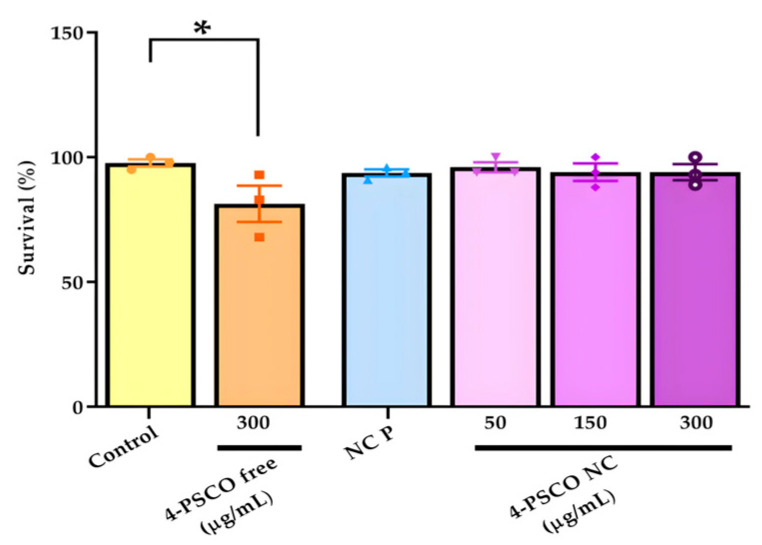
Survival of young adult worms of the N2 strain exposed in liquid medium for 1 h to 4-PSCO free (300 μg/mL), NC P, and different concentrations of 4-PSCO NC (50, 150, and 300 μg/mL). Three independent experiments were performed. Error bars represent mean ± S.E.M. (∗) *p* < 0.05 compared with the control group (one-way ANOVA followed by Dunnett’s post hoc test).

**Figure 8 pharmaceutics-16-00269-f008:**
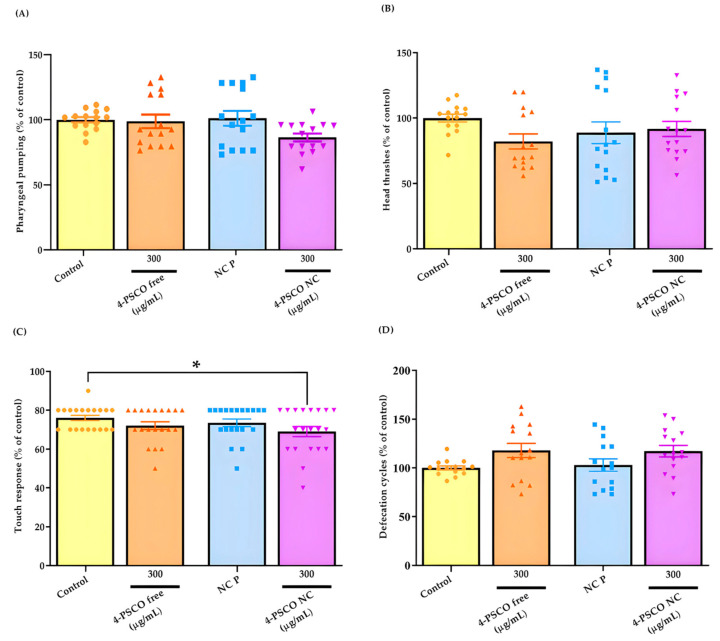
Pharyngeal pumping (**A**), head thrash (**B**), touch response (**C**), and defecation cycle (**D**) assays of young adult worms of the N2 strain exposed in liquid medium for 1 *h* to 4-PSCO free (300 μg/mL), NC P, and 4-PSCO NC (300 μg/mL). Error bars represent mean ± S.E.M. (∗) *p* < 0.05 compared with the control group (one-way ANOVA followed by Dunnett’s post hoc test).

**Figure 9 pharmaceutics-16-00269-f009:**
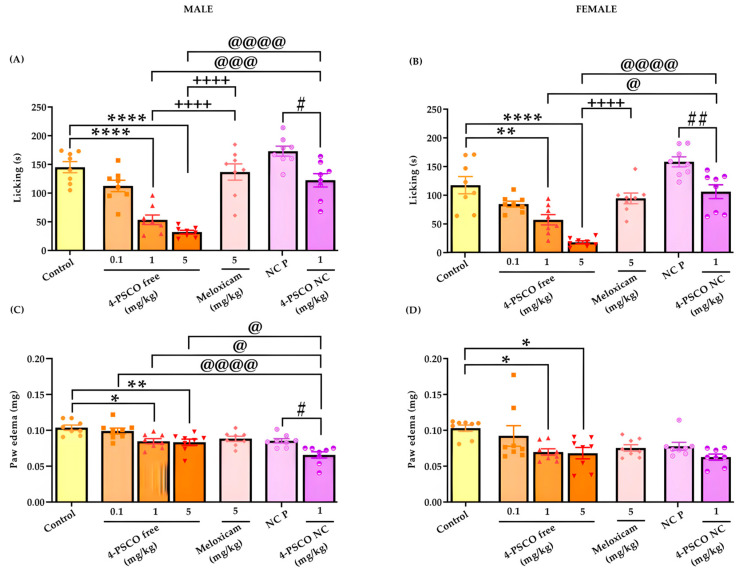
Effect of 4-PSCO free (0.1, 1, and 5 mg/kg), 4-PSCO NC (1 mg/kg), NC P (10 mL/kg), and meloxicam (5 mg/kg) on licking time in male (**A**) and female (**B**) mice; on the edema formation in male (**C**) and female (**D**) mice, glutamate-induced. Each column represents the mean ± S.E.M. of 8 mice in each group. (∗) *p* < 0.05, (∗∗) *p* < 0.01, and (∗∗∗∗) *p* < 0.001 denote significance levels compared with the control group; (#) *p* < 0.05 and (##) *p* < 0.01 denote significance levels compared with the NC P group; (++++) *p* < 0.0001 denotes significance levels compared with the meloxicam group; (@) *p* < 0.05, (@@@) *p* < 0.001, and (@@@@) *p* < 0.0001 denote significance levels compared with the 4-PSCO NC group (one-way ANOVA followed by Tukey’s test).

**Figure 10 pharmaceutics-16-00269-f010:**
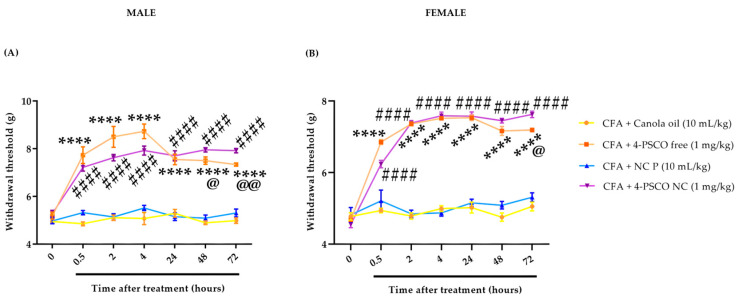
Time–response curve of 4-PSCO free (1 mg/kg), 4-PSCO NC (1 mg/kg), and NC P (10 mL/kg) treatment effect on the mechanical hypernociception induced by Complete Freund’s Adjuvant (CFA) in male (**A**) and female (**B**) mice. The animals received canola oil or NC P (control), 4-PSCO free, or 4-PSCO NC 24 h after CFA injection. The baseline values were recorded before CFA injection. For male mice, the baseline values for canola oil (control), NC P (control), 4-PSCO free, and 4-PSCO NC were 12.52, 12.36, 12.36, and 12.26, respectively. In the case of female mice, the baseline values were 12.76, 12.61, 12.72, and 12.46 for canola oil (control), NC P (control), 4-PSCO free, and 4-PSCO NC, respectively. The measure that followed was 24 h after CFA injection (0) and 0.5, 2, 4, 6, 24, 48, and 72 h after 4-PSCO treatment. Each column represents the mean ± S.E.M. of 8 mice in each group. (∗∗∗∗) *p* < 0. 0.0001 denotes significance levels compared with the control group; (####) *p* < 0.0001 denotes significance levels compared with the NC P group; (@) *p* < 0.05 and (@@) *p* < 0.01 denote significance levels compared with the 4-PSCO NC group (one-way ANOVA followed by Tukey’s test).

**Figure 11 pharmaceutics-16-00269-f011:**
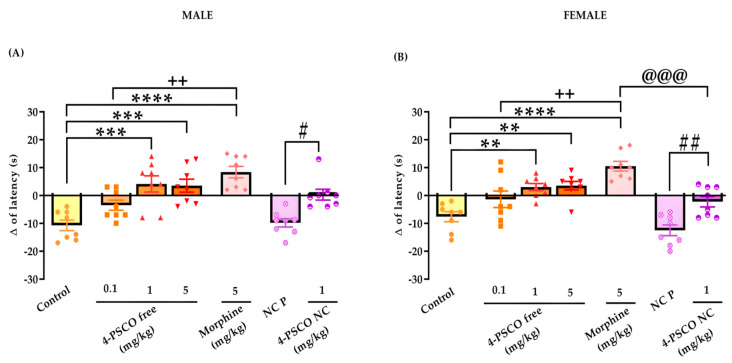
Effect of 4-PSCO free (0.1, 1, and 5 mg/kg), 4-PSCO NC (1 mg/kg), NC P (10 mL/kg), and morphine (5 mg/kg) on response latency to thermal stimulus in the hot plate test in male (**A**) and female (**B**) mice. Each column represents the mean ± S.E.M. of 8 mice in each group. (∗∗) *p* < 0.01, (∗∗∗) *p* < 0.001, and (∗∗∗∗) *p* < 0.0001 denote significance levels compared with the control group; (#) *p* < 0.05 and (##) *p* < 0.01 denote significance levels compared with the NC P group; (++) *p* < 0.01 denotes significance levels compared with the morphine group; (@@@) *p* < 0.001 denotes significance levels compared with the 4-PSCO NC group (one-way ANOVA followed by Tukey’s test).

**Table 1 pharmaceutics-16-00269-t001:** Δ_G_ values (kcal/mol) for 4-PSCO compound, reference drugs, and different proteins.

	MAPK	PAD4	PI3K	JAK2	TLR4	NFkB
**4-PSCO**	−8.3	−6.7	−7.7	−7.9	−7.8	−5.9
**Ketoprofen**	−8.6	–	–	–	–	–
**Naproxen**	–	–	−7.7	−6.3	−7.9	−5.7
**Tofacitinib**	–	−8.8	–	–	–	–
**Dexamethasone**	–	–	−7.8	–	–	–
**Methotrexate**	–	–	–	−8.2	−8.2	−6.6

**Table 2 pharmaceutics-16-00269-t002:** Physicochemical parameters of the 4-PSCO-loaded nanocapsule suspensions (4-PSCO NC) and their respective unloaded (NC P) formulations. The values are reported as mean ± S.E.M. of 3 formulations. The data were analyzed using a Student’s *t*-test (individual times) and one-way ANOVA of repeated measures (stability evaluations) with *p* > 0.05.

**Parameters**					
Samples	Size (nm)	PDI ^a^	ZP ^b^ (mV)	pH	4-PSCO content (%)
**Initial time**					
4-PSCO NC	164 ± 1	0.11 ± 0.01	−18.1 ± 0.4	5.0 ± 0.0	102.4 ± 4.0
NC P	156 ± 2	0.10 ± 0.00	−21.4 ± 0.4	5.4 ± 0.2	-
**15 days**					
4-PSCO NC	136 ± 3	0.10 ± 0.00	−16.4 ± 1.1	5.5 ± 0.0	102.9 ± 4.0
NC P	133 ± 2	0.10 ± 0.00	−20.4 ± 3.9	5.6 ± 0.0	-
**30 days**					
4-PSCO NC	143 ± 2	0.12 ± 0.02	−18.9 ± 0.5	5.5 ± 0.0	102.2 ± 4.0
NC P	138 ± 1	0.10 ± 0.01	−19.8 ± 2.3	5.6 ± 0.1	-

^a^ PDI: polydispersity index; ^b^ ZP: zeta potential.

## Data Availability

Data are contained within the article and Appendix A.

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
