# Peer review of "4-(Phenylselanyl)-2H-chromen-2-one-Loaded Nanocapsule Suspension—A Promising Breakthrough in Pain Management: Comprehensive Molecular Docking, Formulation Design, and Toxicological and Pharmacological Assessments in Mice"

_pharmaceutics, 2024, doi:10.3390/pharmaceutics16020269_

Round 1
Reviewer 1 Report
Comments and Suggestions for Authors
In this paper authors analyzed the effects of 4-(Phenylselanyl)-2H-chromen-2-one (4-PSCO) on pain-associated proteins using molecular docking. I have the following suggestions for authors to improve the manuscript:
1. Some earlier studies examined similar effects of organoselenium compounds in the treatment of pain, so it would be good for the authors to emphasize the importance of their research in relation to previous studies, as well as the advantages and contributions in the treatment of pain:
a) Sousa FSS, Birmann PT, Balaguez R, Alves D, Brüning CA, Savegnago L. α-(phenylselanyl) acetophenone abolishes acute restraint stress induced-comorbid pain, depression and anxiety-related behaviors in mice. Neurochem Int. 2018 Nov;120:112-120. doi: 10.1016/j.neuint.2018.08.006. Epub 2018 Aug 13. PMID: 30114472.
b) Kauane Nayara Bahr Ledebuhr, Gustavo D'Avila Nunes, Evelyn Mianes Besckow, Maira Regina Giehl, Benhur Godoi, Cristiani Folharini Bortolatto, César Augusto Brüning, Antinociceptive effect of N-(3-(phenylselanyl)prop-2-yn-1-yl)benzamide in mice: Involvement of 5-HT1A and 5-HT2A/2C receptors, Chemico-Biological Interactions, Volume 359, 2022, https://doi.org/10.1016/j.cbi.2022.109918.
c) Campos ACP, Antunes GF, Matsumoto M, Pagano RL, Martinez RCR. Neuroinflammation, Pain and Depression: An Overview of the Main Findings. Front Psychol. 2020 Jul 31;11:1825. doi: 10.3389/fpsyg.2020.01825. PMID: 32849076; PMCID: PMC7412934.
d) eis, A.S., Paltian, J.J., Domingues, W.B. et al. Advances in the Understanding of Oxaliplatin-Induced Peripheral Neuropathy in Mice: 7-Chloro-4-(Phenylselanyl) Quinoline as a Promising Therapeutic Agent. Mol Neurobiol 57, 5219–5234 (2020). https://doi.org/10.1007/s12035-020-02048-4.
2. Page 5, Line 189 – missing reference.
3. Page 7, Line 291 - missing reference.
4. Page 9, Table 1 - the authors should highlight which value of binding energy indicate high binding affinity and explain in the disscusion section.
5. Figures and text in all figures are vague. Please increase the resolution of the figures.
6. Figure 9 - When assessing the licking time, was both first response and late response monitored (two distinct periods of high licking activity)?
7. Figure 9 – the authors should try to explain high licking time for NC P.
8. The conclusion is the reputation of abstract. What is the outcome of the study? What about future trends?
Author Response
REVIEWER 1
We appreciate the reviewer for the suggestions and corrections made in the manuscript. We recognize that your comments significantly contribute to improving the quality of the manuscript. The responses to the questions raised are provided sequentially.
Comments: In this paper authors analyzed the effects of 4-(Phenylselanyl)-2H-chromen-2-one (4-PSCO) on pain-associated proteins using molecular docking. I have the following suggestions for authors to improve the manuscript:
Answer: We would like to thank the reviewer for the comments.
Comments: Some earlier studies examined similar effects of organoselenium compounds in the treatment of pain, so it would be good for the authors to emphasize the importance of their research in relation to previous studies, as well as the advantages and contributions in the treatment of pain:
a) Sousa FSS, Birmann PT, Balaguez R, Alves D, Brüning CA, Savegnago L. α-(phenylselanyl) acetophenone abolishes acute restraint stress induced-comorbid pain, depression and anxiety-related behaviors in mice. Neurochem Int. 2018 Nov;120:112-120. doi: 10.1016/j.neuint.2018.08.006. Epub 2018 Aug 13. PMID: 30114472.
b) Kauane Nayara Bahr Ledebuhr, Gustavo D'Avila Nunes, Evelyn Mianes Besckow, Maira Regina Giehl, Benhur Godoi, Cristiani Folharini Bortolatto, César Augusto Brüning, Antinociceptive effect of N-(3-(phenylselanyl)prop-2-yn-1-yl)benzamide in mice: Involvement of 5-HT1A and 5-HT2A/2C receptors, Chemico-Biological Interactions, Volume 359, 2022, https://doi.org/10.1016/j.cbi.2022.109918.
c) Campos ACP, Antunes GF, Matsumoto M, Pagano RL, Martinez RCR. Neuroinflammation, Pain and Depression: An Overview of the Main Findings. Front Psychol. 2020 Jul 31;11:1825. doi: 10.3389/fpsyg.2020.01825. PMID: 32849076; PMCID: PMC7412934.
d) Reis, A.S., Paltian, J.J., Domingues, W.B. et al. Advances in the Understanding of Oxaliplatin-Induced Peripheral Neuropathy in Mice: 7-Chloro-4-(Phenylselanyl) Quinoline as a Promising Therapeutic Agent. Mol Neurobiol 57, 5219–5234 (2020). https://doi.org/10.1007/s12035-020-02048-4.
Answer: We sincerely appreciate the reviewer's insightful comments. Our study contributes to the field of pain management by providing new insights into the effects of a novel organoselenium compound, 4-(Phenylselanyl)-2H-chromen-2-one. While previous studies by Sousa et al. (2018), Ledebuhr et al. (2022), Campos et al. (2020), and Reis et al. (2020) have explored similar effects, our research adds value by elucidating the specific advantages and contributions of our approach. One notable advantage lies in the incorporation of the coumarin nucleus into the molecule. Numerous studies support and demonstrate the advantages and biological effects of these molecules. Although organoselenium compounds have demonstrated efficacy, the inclusion of the coumarin nucleus further substantiates the observed effects. Planar aromatic rings linked with lactone functionality and hydrogen bonds play a crucial role in protein-ligand interactions, enhancing cell recognition and desired pharmacological effects [1]. Structural modifications at positions C-3, C-4, and C-7 further amplify the biological activities of synthetic coumarins [2].
Moreover, while some cited studies sought to elucidate the antinociceptive effects of organoselenium compounds at higher doses (10 mg/kg), in our study we opted for the dose of 1 mg/kg. This choice provides a significant advantage, particularly in terms of mitigating adverse effects and toxicity observed in other organoselenium compounds. Here, compound 4-PSCO did not induce toxicity in the evaluated experimental models, thereby maintaining unchanged behavioral patterns.
Additionally, the studies highlighted by the reviewer focus on investigating the influence of comorbidities associated with pain, such as anxiety and depression, as well as the impact of chronic pain on patients' quality of life. While assessing these factors is indeed relevant, our study aims to evaluate initially a new molecule, specifically for the treatment of acute pain, proposing a novel therapeutic strategy for this condition.
Another advantage observed for 4-PSCO compared to the other compounds mentioned above stems from the development of a formulation for the compound. The development of studies in nanotechnology has been an area of growing interest and scientific research. Nanotechnology offers the ability to manipulate materials at the nanoscale, allowing for the creation of structures and devices with unique and potentially active properties. Additionally, it enables the development of more efficient and targeted drug delivery systems. Nanoparticles, such as nanocapsules, are designed to deliver drugs in a controlled and specific manner, increasing therapeutic efficacy and reducing side effects. Given this, the results of our study enthusiastically show the interesting and promising effects found for this 4-PSCO. To address the reviewer's suggestion, we have included additional information in the discussion section to emphasize the importance of our research to previous studies (Discussion section, page 19, lines 708-719).
References:
- Torres, F.C.; Brucker, N.; Andrade, S.F.; Kawano, D.F.; Garcia, S.C.; Poser, G.L.; Eifler-Lima, V.L. New insights into the chemistry and antioxidant activity of coumarins. Top Med. Chem. 2014, 14, 2600-23. doi: 10.2174/1568026614666141203144551
- Kostova, I. Synthetic and natural coumarins as antioxidants. Mini Rev. Med. Chem. 2006, 6, 365-374. doi: 10.2174/138955706776361457.
Comments: Page 5, Line 189 – missing reference.
Answer: We sincerely appreciate the reviewer for the comment. In response to the reviewer's suggestion, we have included the reference in the revised manuscript (Materials and Methods section, page 5, line 210).
Comments: Page 7, Line 291 - missing reference.
Answer: We thank the reviewer for the comment. The homogenization method employed in sample preparation has been consistently utilized and practiced over the years. Numerous authors have adopted this homogenization protocol, considering it a recommended approach for subsequent biochemical analyses. The following studies cited below utilized this method of sample preparation. To address the reviewer's suggestion, we have incorporated some of the references mentioned below into the revised manuscript (Materials and Methods session, page 8, line 314).
References:
- Klann, I.P.; Martini, F.; Rosa, S.G.; Nogueira, C.W. Ebselen reversed peripheral oxidative stress induced by a mouse model of sporadic Alzheimer's disease. Biol. Rep. 2020, 47, 2205-2215. doi:10.1007/s11033-020-05326-5.
- Fulco, B.C.W.; Jung, J.T.K.; Chagas, P.M.; Rosa, S.G.; Prado, V.C.; Nogueira, C.W. Diphenyl diselenide is as effective as Ebselen in a juvenile rat model of cisplatin-induced nephrotoxicity. Trace Elem. Med. Biol. 2020, 126482.doi:10.1016/j.jtemb.2020.126482.
- Sari, M.H.M.; Ferreira, L.M.; Zborowski, V.A.; Araujo, P.C.O.; Cervi, V.F.; Brüning, C.A.; Cruz, L.; Nogueira, C.W. p,p′-Methoxyl-diphenyl diselenide loaded polymeric nanocapsules are chemically stable and do not induce toxicity in mice. J. Pharm. Biopharm. 2017, 117, 39-48, doi:10.1016/j.ejpb.2017.03.018.
- Birmann, P.T.; Casaril, A.M.; Pesarico, A.P.; Caballero, P.S.; Smaniotto, T.Â.; Rodrigues, R.R.; Moreira, Â.N.; Conceição, F.R.; Sousa, F.S.S.; Collares, T.; Seixas, F.K.; França, R.T.; Corcini, C.D.; Savegnago, L. Komagataella pastoris KM71H modulates neuroimmune and oxidative stress parameters in animal models of depression: A proposal for a new probiotic with antidepressant-like effect. Res. 2021, 171, 105740.doi:10.1016/j.phrs.2021.105740.
- Schossler Garcia, C.; Garcia, P.R.; da Silva Espíndola, C.N.; Nunes, G.D.; Jardim, N.S.; Müller, S.G.; Bortolatto, C.F.; Brüning, C.A. Effect of m-Trifluoromethyl-diphenyl diselenide on the Pain-Depression Dyad Induced by Reserpine: Insights on Oxidative Stress, Apoptotic, and Glucocorticoid Receptor Modulation. Neurobiol. 2021, 58, 5078-5089. doi: 10.1007/s12035-021-02483-x.
Comments: Page 9, Table 1 - the authors should highlight which value of binding energy indicate high binding affinity and explain in the disscusion section.
Answer: Thank you for your valuable comment, with which we concur. We have revised the written form for better clarity. Additionally, in the discussion section, we have highlighted the significance of the binding energy values as requested (Discussion section, page 19, lines 720 – 724).
Comments: Figures and text in all figures are vague. Please increase the resolution of the figures.
Answer: We would like to thank the reviewer for the comment. The resolution of the figures was increased.
Comments: Figure 9 - When assessing the licking time, was both first response and late response monitored (two distinct periods of high licking activity)?
Answer: We would like to thank the reviewer for the comment. Glutamate-induced nociception triggers a behavioral pain response characterized by rapid onset and short duration, followed by the development of edema. Therefore, we only evaluated one type of nociceptive pain, according to the methodology described for the glutamate test [1].
References:
- Beirith, A.; Santos, A.R.; Calixto, J.B. Mechanisms underlying the nociception and paw oedema caused by injection of glutamate into the mouse paw. Brain Res. 2002, 924, 219-228, doi:10.1016/s0006-8993(01)03240-1.
Comments: Figure 9 – the authors should try to explain high licking time for NC P.
Answer: We would like to thank the reviewer for the comment. NC P is the placebo formulation and is made up of all components in the formulation, except 4-PSCO, such as ethylcellulose polymer (100 mg), MCT oil (300 mg), Span® 80 (77 mg), Tween® 80 (77 mg), and water (53 mL). The high lick time is a satisfactory and expected result when compared to 4-PSCO NC. Due to the absence of the compound in the formulation, the antinociceptive and anti-edematogenic effects were not observed. In this case, NC P refers to the vehicle of the formulation containing 4-PSCO, similar to the vehicle (Canola oil) of 4-PSCO in its free form. In this work, we chose to carry out two control groups, NC P and the vehicle (Canola oil). The results found here are in line with other studies, which also demonstrated the absence of the effect of placebo nanoparticles. Villalba et al. [1] demonstrated a high licking time for the control group (suspension without drugs (NC B or NC P)) in glutamate-induced nociception and a significant increase in the formation of paw edema. Pinto et al. [2] also showed that carrageenan injection induced marked and long-lasting paw edema and placebo nanocapsules did not demonstrate an anti-edematogenic effect, as expected.
References:
- Villalba, B.T.; Ianiski, F.R.; Wilhelm, E.A.; Fernandes, R.S.; Alves, M.P.; Luchese, C. Meloxicam-loaded nanocapsules have antinociceptive and antiedematogenic effects in acute models of nociception. Life Sci. 2014, 115, 36-43, doi:10.1016/j.lfs.2014.09.002.
- Pinto, E.P.; Da Costa, S.O.A.M.; D'haese, C.; Nysten, B.; Machado, F.P.; Rocha, L.M.; De Souza, T.M.; Beloqui, A.; Machado, R.R. Araújo, R.S. Poly-É›-caprolactone nanocapsules loaded with copaiba essential oil reduce inflammation and pain in mice. J. Pharm. 2023, 642, 123147, doi:10.1016/j.ijpharm.2023.123147.
Comments: The conclusion is the reputation of abstract. What is the outcome of the study? What about future trends?
Answer: We thank the reviewer for the comments. The study reveals the pharmacological potential of 4-(phenylselenyl)-2H-chromen-2-one (4-PSCO) as a therapeutic agent for pain and inflammation. Computational molecular docking tests have demonstrated a high affinity of 4-PSCO for specific receptors associated with pain, suggesting its potential efficacy in modulating pain pathways. Additionally, the development of a new pharmaceutical formulation based on polymeric nanocapsules has allowed for enhanced delivery and prolonged therapeutic effects of 4-PSCO. Evaluation of 4-PSCO toxicity using Caenorhabditis elegans and Swiss mice did not reveal adverse effects, suggesting its safety profile. Furthermore, both the free and nanoencapsulated forms of 4-PSCO demonstrated antinociceptive, anti-inflammatory, and anti-edematogenic effects in various experimental pain models, including tests with glutamate, hot plate, and a Complete Freund's Adjuvant (CFA)-induced inflammatory pain model.
Notably, the nanoencapsulated form exhibited a more sustained effect, indicating its potential for long-term pain and inflammation management. Moreover, the study emphasizes the importance of exploring selenium-functionalized coumarin compounds, an area of scarce research, as well as the pharmaceutical technology field, which is challenging yet promising, offering considerable potential for the development of safer and more effective therapies. Briefly, the result of the study was to reveal the pharmacological effects and absence of toxicity of a new organic molecule, selenocoumarin, in its free and nanoencapsulated form as a treatment approach for pain and inflammation.
Given the promising and satisfactory results found in this initial study, we aim to deepen our understanding of the mechanisms and risk factors associated with pain in rheumatoid arthritis (RA), as well as the favorable pharmacological effects exerted by the free and nanoencapsulated compound 4-PSCO in this context. The experimental design for future research encompasses the exploration of mechanical and thermal hypersensitivity induced by CFA, along with the investigation of comorbidities linked with RA, such as cognitive deficits and emotional disorders.
Additionally, our objective is to elucidate the involvement of the inflammatory process and oxidative stress in this condition. Consequently, analyze inflammatory parameters, including pro-inflammatory cytokines and proteins involved in signaling pathways, will be conducted using Western Blotting technique. Histological analysis of paw tissues will complement these investigations.
We emphasize our focus on exploring and highlighting the nanoencapsulated form of the compound, recognizing nanotechnology as a promising and rapidly growing field. This pharmaceutical technology enables the enhancement of the biological effects of organic molecules while facilitating modifications aimed at reducing toxicity and adverse effects. Moving forward, our research remains dedicated to advancing innovative pharmacological therapies for pain, inflammation, and associated comorbidities, building on our initial findings. To attend to the reviewer´s suggestion, we have addressed some information about future trends in the revised version (Conclusion session, pages 22, lines 867-882).
Reviewer 2 Report
Comments and Suggestions for Authors
The manuscript entitled "4-(Phenylselanyl)-2H-chromen-2-one-loaded Nanocapsule Suspension - A Promising Breakthrough in Pain Management: Comprehensive Molecular Docking, Formulation Design, Toxicological and Pharmacological Assessments in Mice" by Caren Aline Ramson da Fonseca et al. reports the in sillico assessment of 4-PSCO on the pain management related enzymes. The authors further developed a nano formulation of 4-PSCO and tested their stability, toxicity, in vitro and in vivo effect via i.g. administration.
The molecular docking confirmed the binding of 4-PSCO to the related protein enzymes. Characterization of 4-PSCO NP demonstrated the stability and slow-release profile of this delivery platform. In vivo study exhibits the potential of 4-PSCO on pain management. However, several concerns regarding the 4-PSCO and the study design need to be addressed.
1. The recent review article “Toxicology and pharmacology of synthetic organoselenium compounds: an update” summarizes the current effort of organoselenium on Antinociceptive activity and pain management. In this paper, several organoselenium has been listed as tested for Antinociceptive activity, what is the advantage of 4-PSCO over these compounds? Similar molecules need to be tested either in silico or in experiments as a positive control to demonstrate the benefit of 4-PSCO over other candidates. The novelty of this compound and formulation needs to be addressed.
2. In section 2.7, authors used 1M potassium phosphate buffer at pH 7.4 and 30 % of ethanol as release medium, the rational of this formulation need to be stated. In figure 6, the release profile shown 4-PSCO NC has a slow-release curve. The time point should be at lease 72 hours as 1). The therapeutic effect lasts at least 72 hours. 2). Whether the 4-PSCO could be 100% released needs to be tested.
3. In section 3.2, DLS data shows the 4-PSCO NC is stable at 25C degree for 30 days. It would be interesting to see the stability of formulation at 37C degree. Combining the DLS data and in vitro release curve, we can have a better understanding of the release profile of 4-PSCO NC, whether the release is based on slow diffusion or the dissociation of NC formulation.
4. In section 3.5 and Figure 9A and B, meloxicam as a positive control has been included in the test. However, we didn’t see significant change in licking time comparing control group and Meloxicam group. It is hard to interpret the data without a working possible control group. Also, the NC P group has a significantly higher licking time than control group. What are the reasons for this result? Besides, 4-PSCO NC group also needs to be compared with control group to demonstrate the potential therapeutic effect.
5. In figure 10, The withdrawal threshold has achieved maximum and remains plateau after 2 hours for both 4-PSCO and 4-PSCO NC group. In the previous release profile, at 2 hours less than 10 % 4-PSCO has been released from the NC formulation. With the release time going on, the therapeutic effect didn’t change, what is the reason for this? Is 10 % release enough for the max effect?
6. Line 673, “Acute treatment with 4-PSCO free (1 and 5 mg/kg) in male mice decreased the response latency to the thermal stimulus by 61% and 67% compared with the control, respectively” and Line 678, “At doses of 1 and 5 mg/kg, 4-PSCO free increased the latency time to the thermal stimulus response compared with the control group.” The statement of increase and decrease is not consistent, please correct it accordingly.
7. In Discussion line 807, author mentioned the sustained release profile and broader biodistribution are potential benefit of 4-PSCO NC formulation, even though no significant difference in antinociceptive action were found between 4-PSCO NC and free compound. To test this statement, the blood drug concentration-time curve needs to be studied to see whether the formulation enhances the bioavailability of 4-PSCO, for example, a higher blood concentration or a longer duration time. Without a PK study, it is hard to claim the benefit of 4-PSCO formulation over free drug, especially if the therapeutic effect is the same or lower of 4-PSCO NC than free drugs.
Author Response
REVIEWER 2
We appreciate the reviewer for the suggestions and corrections made in the manuscript. We recognize that your comments significantly contribute to improving the quality of the manuscript. The responses to the questions raised are provided sequentially.
Comments: The manuscript entitled "4-(Phenylselanyl)-2H-chromen-2-one-loaded Nanocapsule Suspension - A Promising Breakthrough in Pain Management: Comprehensive Molecular Docking, Formulation Design, Toxicological and Pharmacological Assessments in Mice" by Caren Aline Ramson da Fonseca et al. reports the in sillico assessment of 4-PSCO on the pain management related enzymes. The authors further developed a nano formulation of 4-PSCO and tested their stability, toxicity, in vitro and in vivo effect via i.g. administration. The molecular docking confirmed the binding of 4-PSCO to the related protein enzymes. Characterization of 4-PSCO NP demonstrated the stability and slow-release profile of this delivery platform. In vivo study exhibits the potential of 4-PSCO on pain management. However, several concerns regarding the 4-PSCO and the study design need to be addressed.
Answer: We thank the reviewer for the comments.
Comments - The recent review article “Toxicology and pharmacology of synthetic organoselenium compounds: an update” summarizes the current effort of organoselenium on Antinociceptive activity and pain management. In this paper, several organoselenium has been listed as tested for Antinociceptive activity, what is the advantage of 4-PSCO over these compounds? Similar molecules need to be tested either in silico or in experiments as a positive control to demonstrate the benefit of 4-PSCO over other candidates. The novelty of this compound and formulation needs to be addressed.
Answer: We would like to thank the reviewer for the comments. The antinociceptive activity of organoselenium compounds is widely recognized, and numerous studies highlight the potential effects of these compounds [1-5]. However, coumarin-derived compounds have garnered significant interest in pharmaceutical research due to their bioactive properties. A notable advantage of these compounds is their ability to interact with multiple biological targets related to pain and inflammation, which may lead to a more comprehensive approach to treating these conditions. Studies demonstrated the potential of coumarin derivatives in modulating inflammatory and nociceptive pathways, thus offering a promising and multifaceted therapeutic strategy [6-9].
Another advantageous aspect of coumarin compounds is their versatile chemical structure, allowing for structural modifications to optimize their pharmacological properties and reduce potential adverse effects. Indeed, planar aromatic rings linked to lactone functionality and hydrogen bonds play a crucial role in protein-ligand interactions, enhancing cellular recognition and desired pharmacological effects [10]. Structural modifications at positions C-3, C-4, and C-7 further amplify the biological activities of synthetic coumarins [11].
Thus, one of the most significant advantages of seleno-coumarin compounds, such as 4-PSCO, lies in their unique chemical structure, combining the beneficial pharmacological properties of both selenium and coumarins. This structural combination allows seleno-coumarin compounds to exhibit antioxidant and antitumor activities, as evidenced in preclinical studies. For example, research conducted by Padilha et al. [12] demonstrated that seleno-coumarin compounds possess potent antioxidant activity, which may significantly contribute to reducing oxidative stress associated with pain and inflammation.
Studies by Lagunes et al. [13] investigated the antiproliferative activity of seleno-coumarins in tumor cell lines, demonstrating potent inhibition of cell proliferation with high selectivity for cancer cells. This enhanced selectivity may reduce the undesirable side effects associated with cancer and conventional drug treatment used. Arsenyan et al. [14] and Domracheva et al. [15] reported the antioxidant and anticancer activities of seleno-coumarin isomers. Recently, Yildirim et al. [16] investigated the pharmacological effects of 3-acetyl coumarin-selenophene in DU-145 tumor cells, as well as the activities of these compounds in apoptosis and oxidative stress in the same cell line.
In light of this, we agree with the reviewer regarding testing similar molecules in silico or positive control. Although additional tests are relevant, our study encompasses a set of results that allows us to infer that 4-PSCO may act in modulating pathways involved in inflammatory and painful processes. This assertion arises from the compound's high binding affinity with different proteins involved in these conditions, as well as the similarity found in the essential amino acid residues for biological effects. Despite not having conducted this comparison, the scientific reports addressed for this class of seleno-coumarin compounds led us to a better understanding of the pharmacological effects of 4-PSCO, making it possible to highlight the findings of the study. To the best of our knowledge, this is the first study to demonstrate the therapeutic effects of seleno-coumarins in the treatment of pain and inflammation in both their free and nanoencapsulated forms, further underscoring the promising advantages of our study.
References:
- Birmann, P.T.; Sousa, F.S.S.; de Oliveira, D.H.; Domingues, M.; Vieira, B.M.; Lenardão, E.J.; Savegnago, L. 3-(4-Chlorophenylselanyl)-1-methyl-1H-indole, a new selenium compound elicits an antinociceptive and anti-inflammatory effect in mice. J. Pharmacol. 2018, 827, 71-79. doi:10.1016/j.ejphar.2018.03.005.
- Rosa, S.G.; Brüning, C.A.; Pesarico, A.P; Souza, A.C.G.; Nogueira, C.W. Anti-inflammatory and antinociceptive effects of 2,2`-dipyridyl diselenide through reduction of inducible nitric oxide synthase, nuclear factor-kappa B and c-Jun N-terminal kinase phosphorylation levels in the mouse spinal cord. Trace Elem. Med. Biol. 2018, 48, 38-45. doi:10.1016/j.jtemb.2018.02.021.
- Wilhelm, E.A.; Soares, P.S.; Reis, A.S.; Barth, A.; Freitas, B.G.; Motta, K.P.; Lemos, B.B.; Vogt, A.G.; da Fonseca, C.A.R.; Araujo, D.R.; Barcellos, A.M.; Perin, G.; Luchese, C.Se — [(2,2-Dimethyl-1,3-dioxolan-4-yl) methyl] 4-chlorobenzoselenolate reduces the nociceptive and edematogenic response by chemical noxious stimuli in mice: Implications of multi-target actions. Rep. 2019, 71, 1201–1209. doi:10.1016/j.pharep.2019.07.003.
- Sacramento, M.; Reis, A.S.; Martins, C.C.; Luchese, C.; Wilhelm, E.A.; Alves, D. Synthesis and Evaluation of Antioxidant, Anti-Edematogenic and Antinociceptive Properties of Selenium-Sulfa Compounds. ChemMedChem. 2022, 17, e202100507. doi: 10.1002/cmdc.202100507.
- Ledebuhr, K.N.B.; Nunes, G.D.; Besckow, E.M.; Giehl, M.R.; Godoi, B.; Bortolatto, C.F.; Brüning, C.A. Antinociceptive effect of N-(3-(phenylselanyl)prop-2-yn-1-yl)benzamide in mice: Involvement of 5-HT1A and 5-HT2A/2C receptors. Biol. Interact. 2022, 359, 109918, doi:10.1016/j.cbi.2022.109918.
- Alipour, M.; Khoobi, M.; Emami, S.; Fallah-Benakohal, S.; Ghasemi-Niri, S.F.; Abdollahi, M.; Foroumadi, A.; Shafiee, A. Antinociceptive properties of new coumarin derivatives bearing substituted 3,4-dihydro-2H-benzothiazines. Daru. 2014, 22, 9. doi: 10.1186/2008-2231-22-9
- Cheriyan, B.V.; Sr., Kadhirvelu P.; Sr., Nadipelly J. Jr.; Shanmugasundaram, J.; Sayeli, V. Sr.; Subramanian, V. Sr. Anti-nociceptive Effect of 7-methoxy Coumarin from Eupatorium Triplinerve vahl (Asteraceae). Mag. 2017, 13, 81-84. doi: 10.4103/0973-1296.197650.
- Alshibl, H.M.; Al-Abdullah, E.S.; Haiba, M.E.; Alkahtani, H.M.; Awad, G.E.A.; Mahmoud, A.H.; Ibrahim, B.M.M.; Bari, A.; Villinger, A. Synthesis and Evaluation of New Coumarin Derivatives as Antioxidant, Antimicrobial, and Anti-Inflammatory Agents. Molecules. 2020, 25, 3251. doi:10.3390/molecules25143251
- Rostom, B.; Karaky, R.; Kassab, I.; Sylla-Iyarreta Veitía, M. Coumarins derivatives and inflammation: Review of their effects on the inflammatory signaling pathways. J. Pharmacol. 2022, 922, 174867. doi:10.1016/j.ejphar.2022.174867.
- Torres, F.C.; Brucker, N.; Andrade, S.F.; Kawano, D.F.; Garcia, S.C.; Poser, G.L.; Eifler-Lima, V.L. New insights into the chemistry and antioxidant activity of coumarins. Top Med. Chem. 2014, 14, 2600-23. doi: 10.2174/1568026614666141203144551
- Kostova, I. Synthetic and natural coumarins as antioxidants. Mini Rev. Med. Chem. 2006, 6, 365-374. doi: 10.2174/138955706776361457.
- Padilha, G.; Birmann,T.; Domingues, M.; Kaufman, T.S.; Savegnago, L.; Silveira, C.C. Convenient Michael addition/β-elimination approach to the synthesis of 4-benzyl- and 4-aryl-selenyl coumarins using diselenides as selenium sources. Tetrahedron Lett. 2017, 58, 985-990, doi:10.1016/j.tetlet. 2017.01.084.
- Lagunes, I.; Begines, P.; Silva, A.; Galán, A.R.; Puerta, A.; Fernandes, M.X.; Maya, I.; Fernández-Bolaños, J.G.; López, Ó.; Padrón, J.M. Selenocoumarins as new multitarget antiproliferative agents: Synthesis, biological evaluation and in silico calculations. J. Med. Chem.2019, 179, 493-501. doi: 10.1016/j.ejmech.2019.06.073.
- Arsenyan, P.; Vasiljeva, J.; Shestakova, I.; Domracheva, I.; Jaschenko, E.; Romanchikova, N.; Leonchiks, A.; Rudevica, Z.; Belyakov, S. Selenopheno[3,2-c]- and [2,3-c]coumarins: Synthesis, cytotoxicity, angiogenesis inhibition, and antioxidant properties. R. Chim. 2015, 18, 399-409. doi.org/10.1016/j.crci.2014.09.007.
- Domracheva, I.; Kanepe-Lapsa, I.; Jackevica, L.; Vasiljeva, J.; Arsenyan, P. Selenopheno quinolinones and coumarins promote cancer cell apoptosis by ROS depletion and caspase-7 activation. Life Sci. 2017, 186, 92-101. doi:10.1016/j.lfs.2017.08.011.
- Yildirim, M.; Ersatir, M.; Arslan, B.; Gİray, E.S. Cytotoxic and apoptotic potential of some coumarin and 2-amino-3-carbonitrile selenophene derivatives in prostate cancer. Turk J. Chem. 2021, 45, 192-198. doi:10.3906/kim-2008-56.
Comments: In section 2.7, authors used 1M potassium phosphate buffer at pH 7.4 and 30 % of ethanol as release medium, the rational of this formulation need to be stated. In figure 6, the release profile shown 4-PSCO NC has a slow-release curve. The time point should be at lease 72 hours as 1). The therapeutic effect lasts at least 72 hours. 2). Whether the 4-PSCO could be 100% released needs to be tested.
Answer: We thank the reviewer for the comment. During our initial studies, we found no release of 4-PSCO when the release medium was comprised solely of 1 M potassium phosphate buffer at pH 7.4. However, we observed that the addition of 30% ethanol to the buffer aids in the concentration gradient of the compound towards the other side of the dialysis membrane. Adding ethanol to the release medium can make it less similar to the physiological medium. However, it helps to reach the sink condition required for this type of evaluation. It's important to note that the in vitro release experiment has limitations and is not a good simulator of in vivo release. This assessment is typically used to evaluate the ability of nanocarriers to release nanoencapsulated compounds and the impact of nanoencapsulation on their release profiles. It can also help predict the release mechanism of the compounds. About the experimental time, after 48 h, the release of the compound from the nanocapsules tends to reach a plateau. Although there were some limitations to the experiment, and the time of the experiment was different than that of in vivo evaluation, there is no doubt that the compound was released in vivo, which resulted in a different effect due to its nanoencapsulation.
Comments: In section 3.2, DLS data shows the 4-PSCO NC is stable at 25C degree for 30 days. It would be interesting to see the stability of formulation at 37C degree. Combining the DLS data and in vitro release curve, we can have a better understanding of the release profile of 4-PSCO NC, whether the release is based on slow diffusion or the dissociation of NC formulation.
Answer: We thank the reviewer for the comment. A stability study was carried out at a temperature of 25°C, taking into account the storage conditions of the formulation. This temperature is recommended by official stability study guidelines. It's important to note that ethylcellulose, which is a slow-dissolving polymer, is not affected by temperature fluctuations. In general, drugs are released from ethylcellulose nanoparticles through diffusion via their polymeric network.
Comments: In section 3.5 and Figure 9A and B, meloxicam as a positive control has been included in the test. However, we didn’t see significant change in licking time comparing control group and Meloxicam group. It is hard to interpret the data without a working possible control group. Also, the NC P group has a significantly higher licking time than control group. What are the reasons for this result? Besides, 4-PSCO NC group also needs to be compared with control group to demonstrate the potential therapeutic effect.
Answer: We appreciate the reviewer for the comments. The insignificance observed for meloxicam may be attributed to the low dose tested. As we aimed to compare the same dose of the positive control and the compound, we administered a dose of 5 mg/kg. Unfortunately, we were unaware of the lack of pharmacological effect of meloxicam at this dosage. However, Sousa et al [1] and Villalba et al. [2] also showed that meloxicam was ineffective in reducing paw edema. Thus, the results of our study are in line with those found in the literature and show that while it is not feasible to directly compare the positive control with 4-PSCO, it is crucial to underscore the promising effects observed for this compound, both in its free form and when nanoencapsulated at low doses.
The significant licking time noted for NC P was expected, given that this formulation serves as the vehicle for 4-PSCO NC. Consequently, NC P comprises all constituents of the formulation, except the compound. Hence, antinociceptive and anti-edematogenic effects were not observed. The vehicle (canola oil) and NC P groups serve as control groups for 4-PSCO free and 4-PSCO NC, respectively. In the male and female mice, no statistical difference was observed in the licking time and paw edema between these two control groups (Vehicle and NC P). For greater clarity and understanding, this sentence has been added to the revised manuscript (Results session, page 16, lines 612-614).
Regarding the comparison between 4-PSCO NC and the vehicle, no significant differences were noted. Nevertheless, the insignificance observed between these experimental groups may stem from differences in the compositions of the administered drugs, potentially altering their mechanisms of action. Therefore, in this study, we assessed two control groups—one for the free compound and another for nanoencapsulated compound—to ensure a comprehensive evaluation of their effects.
References:
- Sousa, F.S.S.; Anversa, R.G.; Birmann, P.T.; de Souza, M.N.; Balaguez, R.; Alves, D., Luchese, C.; Wilhelm, E.A.; Savegnago, L. Contribution of dopaminergic and noradrenergic systems in the antinociceptive effect of α-(phenylalanyl) acetophenone. Rep. 2017, 69, 871–877. doi:10.1016/j.pharep.2017.03.016.
- Villalba, B.T.; Ianiski, F.R.; Wilhelm, E.A.; Fernandes, R.S.; Alves, M.P.; Luchese, C. Meloxicam-loaded nanocapsules have antinociceptive and antiedematogenic effects in acute models of nociception. Life Sci. 2014, 115, 36-43, doi:10.1016/j.lfs.2014.09.002.
Comments: In figure 10, The withdrawal threshold has achieved maximum and remains plateau after 2 hours for both 4-PSCO and 4-PSCO NC group. In the previous release profile, at 2 hours less than 10 % 4-PSCO has been released from the NC formulation. With the release time going on, the therapeutic effect didn’t change, what is the reason for this? Is 10 % release enough for the max effect?
Answer: We thank the reviewer for the comments. As previously stated, the in vitro release test does not fully replicate the physiological conditions of living organisms. Nevertheless, this test is crucial for ensuring the safety and efficacy of the compounds under investigation. Over time, the therapeutic effect of 4-PSCO does not diminish, given the slow release of the compound from the formulation, as evidenced by the in vitro release profile.
Upon evaluating the other times for both release kinetics and the time-response curve, no significant differences or abrupt peaks were detected between the times, confirming the sustained pharmacological effect. As the reviewer mentioned, administration of the compound after 2 hours resulted in a reduction in mechanical hypersensitivity, with this effect lasting up to 72 hours in male and female mice.
However, a 10% release is insufficient to achieve the maximum effect, as the most significant effects were observed at 48 and 72 hours compared to free 4-PSCO. Notably, after 24 and 48 hours, the release curve begins to display more pronounced changes, releasing a larger quantity of 4-PSCO. This observation is supported by a decrease in hypersensitivity in the animals, with the 48 and 72-hour time points showing the most promising results.
Comments: Line 673, “Acute treatment with 4-PSCO free (1 and 5 mg/kg) in male mice decreased the response latency to the thermal stimulus by 61% and 67% compared with the control, respectively” and Line 678, “At doses of 1 and 5 mg/kg, 4-PSCO free increased the latency time to the thermal stimulus response compared with the control group.” The statement of increase and decrease is not consistent, please correct it accordingly.
Answer: We thank the reviewer for their comment, with which we agree. It was a mistake. The mentioned results have been improved and corrected accordingly (Results session, pages 17 and 18, lines 662, 663, and 675).
Comments: In Discussion line 807, author mentioned the sustained release profile and broader biodistribution are potential benefit of 4-PSCO NC formulation, even though no significant difference in antinociceptive action were found between 4-PSCO NC and free compound. To test this statement, the blood drug concentration-time curve needs to be studied to see whether the formulation enhances the bioavailability of 4-PSCO, for example, a higher blood concentration or a longer duration time. Without a PK study, it is hard to claim the benefit of 4-PSCO formulation over free drug, especially if the therapeutic effect is the same or lower of 4-PSCO NC than free drugs.
Answer: We agree with the reviewer's comments, and we acknowledge the importance and relevance of studying the pharmacokinetic profile of 4-PSCO. Testing a concentration-time curve would be crucial to evaluate the therapeutic effect of the compound. However, at the moment, we do not have the resources to conduct this assay, and for this reason, we acknowledge it as one of the limitations of our research. In the present study, statistically significant differences between free 4-PSCO and 4-PSCO NC were identified. However, we expected the statistical difference to be greater between dosage forms. The distinctions were noted at 48 and 72 hours for male mice and at 72 hours for female mice. Here, it is important to highlight that factors such as the sustained release profile, broader biodistribution of 4-PSCO resulting in a higher cellular concentration, favorable solubility within the formulation matrix, and the diminutive size of the suspended nanoparticles substantiate the observed delayed disparities between the 4-PSCO free and 4-PSCO NC. In response to the reviewer's suggestion and for greater clarity and understanding of the study results, the initial sentence of the paragraph was rewritten, and additional information was added in the discussion section. These changes also acknowledge the limitations of our study (Discussion section, page 21 and 22, lines 843-846 and 849-851).
Reviewer 3 Report
Comments and Suggestions for Authors
The study reports 4-PSCO as a new drug candidate for pain management. Although, various aspects of 4-PSCO were investigated, there are weak points in this study where modifications and/ore more precise information would be necessary for the evaluation of the novelty, soundness and significance of this study.
1) Introduction. It is unclear how 4-PSCO was discovered. Are there other studies which reported similar effects of this compound previously? Without such information, it is impossible to judge the novelty of this compound (or the novelty of the use of this compound against pain and inflammation).
2) An overview of (similar) compounds used against pain and inflammation is also missing from the Introduction.
3) The docking results (energies, interaction pattern) of 4-PSCO should be compared to the docking results of other compounds of proven activity (references). This has to be done for each target protein. Without such comparisons to reference compounds, it is impossible to see if 4-PSCO is a good binder to a target or not.
4) Experimental binding assays of 4-PSCO to at least some of the listed drug targets has to be involved to verify computational docking predictions. The same references can be used as in point 3.
5) Some of the reference compounds (point 3) should be used in the in vivo experiments, as well.
Author Response
REVIEWER 3
We appreciate the reviewer for the suggestions and corrections made in the manuscript. We recognize that your comments significantly contribute to improving the quality of the manuscript. The responses to the questions raised are provided sequentially.
Comments: The study reports 4-PSCO as a new drug candidate for pain management. Although, various aspects of 4-PSCO were investigated, there are weak points in this study where modifications and/ore more precise information would be necessary for the evaluation of the novelty, soundness and significance of this study.
Answer: We thank the reviewer for the comments.
Comments: Introduction. It is unclear how 4-PSCO was discovered. Are there other studies which reported similar effects of this compound previously? Without such information, it is impossible to judge the novelty of this compound (or the novelty of the use of this compound against pain and inflammation).
Answer: We thank the reviewer for the comment, with which we agree. To the best of our knowledge, this study represents the first demonstration of the pharmacological effects of the compound 4-PSCO as a therapeutic strategy for pain and inflammation. Padilha et al. [1] synthesized a class of derivatives of 4-benzyl- and 4-aryl-selenyl coumarins, among which the compound 4-PSCO is included, and evaluated some compounds for their potential antioxidant activity. The results obtained are encouraging, as the tested compounds showed a decrease in the levels of reactive species determined with dichlorodihydrofluorescein-2,7-diacetate (DCHF-DA) starting from a concentration of 0.1 µM in the hippocampus and cerebral cortex of mice.
Additionally, the levels of lipid peroxidation induced by sodium nitroprusside (SNP) were measured using the thiobarbituric acid reactive substances (TBARS) assay. The results revealed that the compound 3-Bromo-4-(ptolylselenyl)-2H-chromen-2-one significantly reduced lipid peroxidation levels at a concentration of 5 µM. Thus, these preliminary findings indicate promising antioxidant properties for this class of novel compounds, further motivating continued research with 4-PSCO [1].
Moreover, another class of similar compounds, namely selenofeno[2,3-c] and -[3,2-c]coumarins substituted, demonstrated antiproliferative activities in various tumor cell lines and exhibited the ability to inhibit metalloproteinase-2 and angiogenesis [2]. Lagunes et al. [3] investigated the antiproliferative effects of three families derived from selenocoumarins, namely isoselenocyanate, selenocarbamates, and selenoureas. The synthesis of antiproliferative agents with a multi-target mode of action was promising, as these compounds exhibited high selectivity for tumor cell lines. Additionally, in silico tests revealed a strong interaction of selenoderivatives with the active site of Histone Deacetylase 8 (HDAC8).
Currently, there is a lack of knowledge regarding compounds similar to selenocoumarins used in the treatment of pain and inflammation. Only organoselenium compounds and coumarin-derived compounds possess these pharmacological properties. Therefore, the findings of this study are relevant, given the promising results observed in the aforementioned studies, indicating a new pharmacological approach for this class of compounds. An overview of the pharmacological effects of seleno-coumarin compounds has been added to the introduction section (Introduction session, page 2, lines 62-76).
References:
- Padilha, G.; Birmann, P.T.; Domingues, M.; Kaufman, T.S.; Savegnago, L.; Silveira, C.C. Convenient Michael addition/β-elimination approach to the synthesis of 4-benzyl- and 4-aryl-selenyl coumarins using diselenides as selenium sources. Tetrahedron Lett. 2017, 58, 985-990, doi:10.1016/j.tetlet. 2017.01.084.
- Arsenyan, P.; Vasiljeva, J.; Shestakova, I.; Domracheva, I.; Jaschenko, E.; Romanchikova, N.; Leonchiks, A.; Rudevica, Z.; Belyakov, S. Selenopheno[3,2-c]- and [2,3-c]coumarins: Synthesis, cytotoxicity, angiogenesis inhibition, and antioxidant properties. R. Chim. 2015, 18, 399-409. doi.org/10.1016/j.crci.2014.09.007.
- Lagunes, I.; Begines, P.; Silva, A.; Galán, A.R.; Puerta, A.; Fernandes, M.X.; Maya, I.; Fernández-Bolaños, J.G.; López, Ó.; Padrón, J.M. Selenocoumarins as new multitarget antiproliferative agents: Synthesis, biological evaluation and in silico calculations. J. Med. Chem.2019, 179, 493-501. doi: 10.1016/j.ejmech.2019.06.073.
Comment: An overview of (similar) compounds used against pain and inflammation is also missing from the Introduction.
Answer: We thank the reviewer for their comment. However, to the best of our knowledge, this study represents the first demonstration of the pharmacological effects of the compound 4-PSCO, a seleno-coumarin compound, as a therapeutic strategy for pain and inflammation. For this reason, we did not find studies that evaluated seleno-coumarin compounds under these conditions. Only coumarin derivatives and organic selenium compounds have demonstrated relevant antinociceptive and anti-inflammatory properties. However, other pharmacological effects of this class of compounds, such as antioxidant and anticancer effects, have been addressed and can be found on page 2, lines 62-76 of the introduction session. The findings of these studies are significant, given the promising results observed in the studies, providing us with more support regarding the effects of 4-PSCO and highlighting its advantages and efficacy.
Comments: The docking results (energies, interaction pattern) of 4-PSCO should be compared to the docking results of other compounds of proven activity (references). This has to be done for each target protein. Without such comparisons to reference compounds, it is impossible to see if 4-PSCO is a good binder to a target or not.
Answer: We appreciate the reviewer's comment and, in response to your request, the following text provides a comparative analysis between reference compounds and 4-PSCO.
Recently, Radu et al. (2023) evaluated the affinity of disease-modifying antirheumatic drugs with Janus Kinase. The drugs tofacitinib, baricitinib, peficitinib, upadacitinib and filgotinib were investigated considering their promising effects in the treatment of rheumatoid arthritis. In our study, 4-PSCO exhibited a binding energy of -7.9 kcal/mol, and π-sigma interactions with residues Leu983, Val863, and Leu855, and π-alkyl interactions with Ala880, Leu932, and Met929. Similarly, the reference drugs interacted with the same amino acid residues and demonstrated a binding energy of -8.3 (Baricitinibe) and -9,5 kcal/mol (Peficitinibe). Here, the similarities in interactions suggest comparable binding affinities for JAK2 and highlight the anti-inflammatory effects of 4-PSCO [1].
Regarding NF-κB, Vardhini et al. [2] investigated the interaction of non-steroidal anti-inflammatory drugs against NF-κB-mediated inflammation. The binding energy values for naproxen, aspirin, ibuprofen, dexamethasone, and tamoxifen were -6.33 kcal/mol, -4.69 kcal/mol, -5.61 kcal/mol, -6.21 kcal/mol, and -6.35 kcal/mol, respectively. In our study, the interaction between 4-PSCO and NF-κB resulted in a value of -5.9 kcal/mol. Based on these findings, it can be inferred that the binding affinities are comparable, suggesting that the compound may effectively modulate this pathway [2].
Additionally, another study investigated the mechanisms of action of imperatorin, a furocoumarin drug, and its interaction with the TLR4 protein. Imperatorin binds to TLR4 co-receptors, inhibiting their binding and subsequent signaling. Imperatorin and dexamethasone used in the study interact with the TLR4 target through similar amino acid residues, such as Leu61, Phe119, and Phe151, which were also observed in the interaction of 4-PSCO with TLR4. The binding energy for 4-PSCO and imperatorin was -7.8 kcal/mol and -8.4 kcal/mol, respectively, highlighting the promising effects of 4-PSCO in treating inflammatory conditions [3].
Still, Kang's study et al. (2020) evaluated the effects of ketoprofen, a non-steroidal anti-inflammatory drug, on proteins involved in its metabolism. Here, we highlight the p38MAPK protein. In Kang's work, ketoprofen showed good affinity with p38MAPK, revealing a binding energy of -6.79 kcal/mol. In our study, the interaction of 4-PSCO with the MAPK protein presented a binding energy of -8.3 kcal/mol. Comparing the reference drug and 4-PSCO, it is clear to observe a greater binding affinity for our study compound. Thus, this high interaction observed between 4-PSCO and p38MAPK highlights the interesting and promising analgesic and anti-inflammatory effects of this compound [4].
Due to the short timeframe for responding to inquiries, we were unable to search the literature for the interaction of reference medications with the PI3K, and PAD4 proteins. However, to highlight the pharmacological effects of 4-PSCO through a quicker search, the articles cited below provide a comparison between coumarin-derived compounds and the compound under study.
The first study investigated the interaction between 4-PSCO and the PAD4 receptor and hydrophobic interactions were observed. Nadzirin et al. [5] investigated the inhibitory effect of α-enolase analog peptides on PAD4 and found promising inhibition with canavanine [P2 (Cav)]. Notably, 4-PSCO contains a carbonyl and a cyclic ether group, similar to the P2 (Cav) peptide. Active residues such as Asp350, Asp473, Arg374, and His640 facilitate peptide binding to the PAD4 receptor, and in our molecular docking analysis of 4-PSCO, Asp374 was found, suggesting its potential anchoring with PAD4 [5].
We also examined the interaction of selenomethionine and γ-glutamyl-methyl-selenocysteine (gamma-GluMetSeCys) with the PI3K target. Both compounds and 4-PSCO exhibited interactions with the PI3K target through amino acids Met804, Ile831, and Trp812. While 4-PSCO demonstrated a binding energy of -7.7 kcal/mol, selenomethionine, and gamma-GluMetSeCys showed values of -4.5 and -5.04 kcal/mol, respectively [6].
Overall, 4-PSCO demonstrated comparable outcomes to reference drugs and the compounds examined by previous researchers. The interaction with key amino acids crucial for molecular binding exhibited consistency between reference drugs and organic compounds with 4-PSCO. We highlight that the interaction between 4-PSCO and MAPK demonstrated a higher affinity of the compound for the protein when compared to the reference drug (ketoprofen). Consequently, the comparative analysis herein elucidates a notable affinity of this compound with proteins implicated in painful and inflammatory conditions, indicating its potential capacity to modulate and efficaciously intervene in the management of such diseases. To address the reviewer's suggestion, the results of molecular docking between reference drugs and the compound 4-PSCO were included in the discussion section of the revised manuscript (Discussion session, pages 19 and 20, lines 731-764).
References:
- Radu, A.F.; Bungau, S.G.; Negru, A.P.; Uivaraseanu, B.; Bogdan, M.A. Novel Potential Janus Kinase Inhibitors with Therapeutic Prospects in Rheumatoid Arthritis Addressed by In Silico Studies. Molecules. 2023, 28, 4699. doi:10.3390/molecules28124699.
- Vardhini, S.P.; Sadiya, H.; Beigh, S.; Pandurangan, A.K.; Srinivasan, H.; Anwer, M.K.; Waseem, M. Possible Interaction of Nonsteroidal Anti-inflammatory Drugs Against NF-κB- and COX-2-Mediated Inflammation: In Silico Probe. Biochem. Biotechnol. 2022, 194, 54-70. doi:10.1007/s12010-021-03719-1.
- Huang, M.H.; Lin, Y.H.; Lyu, P.C.; Liu, Y.C.; Chang, Y.S.; Chung, J.G.; Lin, W.Y.; Hsieh, W.T. Imperatorin Interferes with LPS Binding to the TLR4 Co-Receptor and Activates the Nrf2 Antioxidative Pathway in RAW264.7 Murine Macrophage Cells. Antioxidants (Basel). 2021, 10, 362. doi:10.3390/antiox10030362.
- Kang, N.H.; Mukherjee, S.; Jang, M.H.; Pham, H.G.; Choi, M.; Yun, J.W. Ketoprofen alleviates diet-induced obesity and promotes white fat browning in mice via the activation of COX-2 through mTORC1-p38 signaling pathway. Pflugers Arch. 2020, 472, 583-596. doi:10.1007/s00424-020-02380-7.
- Ahmad Nadzirin, I.; Chor, A.L.T.; Salleh, A.B.; Rahman, M.B.A.; Tejo, B.A. Discovery of new inhibitor for the protein arginine deiminase type 4 (PAD4) by rational design of α-enolase-derived peptides. Biol. Chem. 2021, 92, 107487. doi:10.1016/j.compbiolchem.2021.107487.
- Shalihat, A.; Lesmana, R.; Hasanah, A.N.; Mutakin, M. Selenium Organic Content Prediction in Jengkol (Archidendron pauciflorum) and Its Molecular Interaction with Cardioprotection Receptors PPAR-γ, NF-κB, and PI3K. Molecules. 2023, 28, 3984. doi:10.3390/molecules28103984.
Comments: Experimental binding assays of 4-PSCO to at least some of the listed drug targets has to be involved to verify computational docking predictions. The same references can be used as in point 3.
Answer: We thank the reviewer for the comment. Indeed, analyzing in practice the interaction of 4-PSCO with the proteins selected for molecular docking is extremely interesting. However, in the current study, the selection of proteins for the in silico test was driven by an exploratory approach. This decision stems from the innovative nature of the compound under investigation and our pursuit of novel strategies for pain and inflammation treatment. Initially, our objective was to broaden the scope of investigation and explore the potential mechanisms of action of 4-PSCO.
While it is relevant and important to experimentally evaluate the action of the compound in these pathways, our understanding of the compound's mechanisms of action was limited, hindering the ability to design a more targeted study for selecting just a few proteins. Despite our knowledge of the pharmacological properties of selenium and coumarin derivatives, a comprehensive understanding of 4-PSCO mode of action was lacking. Therefore, we opted to assess the pharmacological effects of the compound across various pain and inflammation models, alongside toxicity evaluation. Thus, based on the results obtained from both the in silico and in vivo tests, further studies are currently underway to address the reviewer's feedback and refine our research focus regarding mechanisms of action of 4-PSCO.
Comment: Some of the reference compounds (point 3) should be used in the in vivo experiments, as well.
Answer: We thank the reviewer for the comment. Indeed, evaluating the pharmacological effects of reference drugs in vivo is extremely important and relevant. Nevertheless, the use of animals in scientific research necessitates meticulous consideration of ethical principles, encompassing the 3Rs principles (Reduction, Refinement, and Substitution), and a robust justification when seeking to augment the number of animals for supplementary tests. We recognize the importance of evaluating the therapeutic effects of molecules in vivo experiments; however, considering the ethical approach that advocates alternative methods whenever feasible and aims to minimize the impact on living organisms, evaluating the pharmacological effects of various reference medications demands a substantial number of animals. Moreover, the assessment of toxicity amplifies the necessity for an increased animal count. Given the substantial results achieved and the extensive use of animals in these experiments, acquiring additional animals for further tests poses a challenge. In this study, we evaluated the biological effects of 4-PSCO in different models of pain and inflammation that have been established, concurrently assessing the compound's toxicity in two distinct models. Therefore, we believe that the set of results obtained from this initial study is satisfactory for publication since additional studies are already being planned. The comparison of the effects observed for the 4-PSCO compound with medications already employed in clinical practice is very interesting and pertinent. Thus, the reviewer's suggestion will be duly noted and implemented in future studies.
Reviewer 4 Report
Comments and Suggestions for Authors
The manuscript submitted for review describes interesting and promising research on potentially analgesic substances and the method of their delivery to the patient's body. Enclosing the active substance in a nanocapsule can reduce acute and systemic toxicity to parenchymal organs. A few substantive remarks come to mind: The introduction does not contain information about the tested chemical compound, its source is found only in the next chapter describing the materials. The reader does not know the reasons why the authors chose the 4-PSCO compound and tested it in the presented tests. It seems advisable for the authors to reveal why this particular compound was selected for research out of many described by the authors in the source [19]. The information contained in lines 105-106 is not exhaustive and certainly does not justify the authors' choice.
The described studies of the 4-PSCO compound were planned and carried out in accordance with generally applicable standards, but they seem to be disconnected from each other. Molecular docking has been described in detail. It would be advisable to supplement computational studies with the affinity of 4-PSCO for cyclooxygenase (cox-1 and cox-2). Since in another study the nociceptive effect of the test substance was compared with a reference drug with preferred properties to inhibit this enzyme, the authors should compare two methods: in vitro methods to in silico methods. The choice of meloxicam as the reference drug in the glutamate test in male and female mice appears unjustified. Meloxicam is not a representative example of a glutamate antagonist. The above thesis is confirmed by the research results. Statistically insignificant activity of meloxicam was found compared to the control sample.
As the authors note, the hot plate test describes analgesic properties with a central, and certainly supraspinal, mechanism, it does not provide information about anti-inflammatory effects, therefore it would be more justified to plan a formalin test, the first phase of which describes the immediate analgesic effect, while the second phase suggests the potential anti-inflammatory properties of the tested substance. The formatting of the manuscript in the template also requires improvement: table 2 is difficult to read and should be placed in such a way that the row numbers are not in the first column. Other charts and texts describing symbols could be more readable (larger).
Author Response
REVIEWER 4
We appreciate the reviewer for the suggestions and corrections made in the manuscript. We recognize that your comments significantly contribute to improving the quality of the manuscript. The responses to the questions raised are provided sequentially.
Comments: The manuscript submitted for review describes interesting and promising research on potentially analgesic substances and the method of their delivery to the patient's body. Enclosing the active substance in a nanocapsule can reduce acute and systemic toxicity to parenchymal organs. A few substantive remarks come to mind: The introduction does not contain information about the tested chemical compound, its source is found only in the next chapter describing the materials. The reader does not know the reasons why the authors chose the 4-PSCO compound and tested it in the presented tests. It seems advisable for the authors to reveal why this particular compound was selected for research out of many described by the authors in the source [19]. The information contained in lines 105-106 is not exhaustive and certainly does not justify the authors' choice.
Answer: We thank the reviewer for the comment. As required, more detailed information about the 4-PSCO compound selection was added to the introduction session of the revised manuscript (Introduction session, page 2, lines 52-76).
Comments: The described studies of the 4-PSCO compound were planned and carried out in accordance with generally applicable standards, but they seem to be disconnected from each other. Molecular docking has been described in detail. It would be advisable to supplement computational studies with the affinity of 4-PSCO for cyclooxygenase (cox-1 and cox-2). Since in another study the nociceptive effect of the test substance was compared with a reference drug with preferred properties to inhibit this enzyme, the authors should compare two methods: in vitro methods to in silico methods. The choice of meloxicam as the reference drug in the glutamate test in male and female mice appears unjustified. Meloxicam is not a representative example of a glutamate antagonist. The above thesis is confirmed by the research results. Statistically insignificant activity of meloxicam was found compared to the control sample.
Answer: We appreciate the reviewer's input and partially agree with their suggestion. Assessing the interaction of 4-PSCO with cyclooxygenases would indeed provide valuable insights into its involvement in the inflammatory process, particularly considering the positive control used in the glutamate test. While these specific proteins were not evaluated in our study, the selection and findings concerning p38 MAP kinase, peptidyl arginine deiminase type 4, phosphoinositide 3-kinase, Janus kinase 2, toll-like receptor 4, and nuclear factor-kappa ? demonstrate promise and offer a broader perspective on the potential interactions and therapeutic effects of the new compound. Since it is an initial study, our primary aim was to enhance our understanding of the mechanisms of action of 4-PSCO by examining various proteins implicated in the treatment of pain and inflammation. Thus, based on the results obtained and considering the reviewer's feedback, we are currently planning further studies to address all suggestions.
Meloxicam, a non-steroidal anti-inflammatory drug, possesses analgesic properties and is known to alleviate pain and inflammation. Given its established pharmacological effects and widespread use in clinical practice and in vivo studies, we selected meloxicam as a positive control in the glutamate-induced nociception test. However, in our study, it is challenging to explain why meloxicam did not significantly reduce licking time and paw edema in the animal’s glutamate-induced. We intended to administer the compound and positive control at the same dose (5 mg/kg) to compare their biological effects. Given this, the study by Souza et al. [1] also showed that meloxicam was ineffective in inhibiting paw edema could be attributed to the low dosage administered. Additionally, Villalba et al. [2] showed that meloxicam-loaded nanocapsules significantly reduced paw edema induced by glutamate, whereas free meloxicam did not exhibit efficacy in this test. Consequently, our findings are consistent with those reported in the literature and underscore the antinociceptive and anti-edematogenic effects of free and nanoencapsulated 4-PSCO in lower dose compared to the dose selected for meloxicam
References:
- Sousa, F.S.S.; Anversa, R.G.; Birmann, P.T.; de Souza, M.N.; Balaguez, R.; Alves, D., Luchese, C.; Wilhelm, E.A.; Savegnago, L. Contribution of dopaminergic and noradrenergic systems in the antinociceptive effect of α-(phenylalanyl) acetophenone. Rep. 2017, 69, 871–877. doi:10.1016/j.pharep.2017.03.016.
- Villalba, B.T.; Ianiski, F.R.; Wilhelm, E.A.; Fernandes, R.S.; Alves, M.P.; Luchese, C. Meloxicam-loaded nanocapsules have antinociceptive and antiedematogenic effects in acute models of nociception. Life Sci. 2014, 115, 36-43, doi:10.1016/j.lfs.2014.09.002.
Comments: As the authors note, the hot plate test describes analgesic properties with a central, and certainly supraspinal, mechanism, it does not provide information about anti-inflammatory effects, therefore it would be more justified to plan a formalin test, the first phase of which describes the immediate analgesic effect, while the second phase suggests the potential anti-inflammatory properties of the tested substance. The formatting of the manuscript in the template also requires improvement: table 2 is difficult to read and should be placed in such a way that the row numbers are not in the first column. Other charts and texts describing symbols could be more readable (larger).
Answer: We appreciate the reviewer's comment. Evaluating the effects of 4-PSCO on the formalin test would be of great interest and would further expand our understanding of the pharmacological effects of 4-PSCO. However, due to the necessity of minimizing the use of animals in experiments, and considering the principles of the 3Rs (Reduction, Refinement, and Replacement), this trial was not conducted. Combined with the principles of the 3Rs, we chose to evaluate the glutamate-induced nociception test and a time-response curve induced by Complete Freund’s Adjuvant (CFA) to assess the antinociceptive, anti-edematogenic, and anti-inflammatory effects of 4-PSCO.
Glutamate, a prominent excitatory neurotransmitter, plays a crucial role in inducing nociception in both the peripheral and spinal nervous systems through the activation of specific receptors. The administration of glutamate in the intraplantar region of the hind paw elicits harmful stimuli, activating synapses in peripheral and central neurons, as well as initiating the synthesis of inflammatory cytokines, leading to the observed reaction in animals [1-2].
The inflammatory pain induction protocol using CFA is also widely employed to investigate the mechanisms associated with inflammation and to assess the anti-inflammatory effects of compounds in acute and chronic pain [3]. The administration of CFA results in the release of several inflammatory mediators, triggering a cascade of biochemical, morphological, and molecular events in the paw, as well as inducing sensitization in specific regions of the central nervous system [4-6]. This experimental model replicates the clinical conditions of human persistent inflammatory pain, making it a reliable model for assessing the pharmacological effects of novel compounds [3].
Thus, based on these literature reports, it is possible to emphasize that the two pain models used in the study effectively and accurately evaluate the antinociceptive, anti-edematogenic, and anti-inflammatory effects of 4-PSCO. However, in future studies, we could conduct additional experiments to evaluate the effects of the compound on the formalin test and provide further insights into the effects of 4-PSCO in managing painful and inflammatory conditions.
Regarding the formatting of the manuscript, all points addressed by the reviewer were improved in the revised manuscript.
References:
- De Souza, M.M.; Pereira, M.A.; Ardenghi, J.V.; Mora, T.C.; Bresciani, L.F.; Yunes, R.A.; Delle Monache, F.; Cechinel-Filho, V. Filicene obtained from Adiantum cuneatum interacts with the cholinergic, dopaminergic, glutamatergic, GABAergic, and tachykinergic systems to exert antinociceptive effect in mice. Biochem. Behav. 2009, 93, 40-46. doi:10.1016/j.pbb.2009.04.004.
- Wang, Z.; Que, B.; Gan, J.; Guo, H.; Chen, Q.; Zheng, L.; Marraiki, N.; Elgorban, A.M.; Zhang, Y. Zinc oxide nanoparticles synthesized from Fraxinus rhynchophylla extract by green route method attenuates the chemical and heat induced neurogenic and inflammatory pain models in mice. Photochem. Photobiol. B. 2020, 202, 111668. doi:10.1016/j.jphotobiol.2019.111668.
- Piegang, B.N.; Ndjateu, F.S.T.; Tene, M.; Bomba, F.D.T.; Tseuguem, P.P.; Nguelefack, T.B. Antinociceptive, anti-inflammatory and antioxidant effects of Boerhavia coccinea extracts and fractions on acute and persistent inflammatory pain models. Basic Clin. Physiol. Pharmacol. 2021, 32, 20200118. doi: 10.1515/jbcpp-2020-0118.
- Fehrenbacher, J.C.; Vasko, M.R.; Duarte, D.B. Models of inflammation: Carrageenan- or complete Freund's Adjuvant (CFA)-induced edema and hypersensitivity in the rat. Protoc. Pharmacol. 2012, 5, Unit5.4. doi:10.1002/0471141755.ph0504s56.
- Billiau, A.; Matthys, P. Modes of action of Freund's adjuvants in experimental models of autoimmune diseases. Leukoc. Biol. 2001, 70, 849-860.
- Araujo, P.C.O.; Sari, M.H.M.; Jardim, N.S.; Jung, J.T.K.; Brüning, C.A. Effect of m-trifluoromethyl-diphenyl diselenide on acute and subchronic animal models of inflammatory pain: Behavioral, biochemical and molecular insights. Biol. Interact. 2020, 317, 108941. doi:10.1016/j.cbi.2020.108941.
Round 2
Reviewer 2 Report
Comments and Suggestions for Authors
Thanks authors for the revision. All my concerns have been addressed.
Author Response
We would like to thank the reviewer once again for their comment.Reviewer 3 Report
Comments and Suggestions for Authors
The requested docking calculations were not performed with reference compounds. (I expected docking calculations using the same methodology for all compounds including references and not literature search.) The requested in vitro binding experiments were not performed. Thus, I cannot judge the significance of the 4-PSCO action compared to other reference compounds, and also cannot confirm its proposed binding to different protein targets.
Author Response
REVIEWER 3
We appreciate the reviewer for the suggestions and corrections made in the manuscript. We recognize that your comments significantly contribute to improving the quality of the manuscript. The responses to the questions raised are provided sequentially.
Comment: The requested docking calculations were not performed with reference compounds. (I expected docking calculations using the same methodology for all compounds including references and not literature search.) The requested in vitro binding experiments were not performed. Thus, I cannot judge the significance of the 4-PSCO action compared to other reference compounds, and also cannot confirm its proposed binding to different protein targets.
Answer: We would like to thank the reviewer once again for their comment. To address your comment, we assessed the interaction between the selected proteins and the reference drugs commonly used to treat pain and inflammation. These drugs include Ketoprofen, Naproxen, Tofacitinib, Dexamethasone, and Methotrexate. The affinity values of the drugs with the target proteins are shown in Table 1 of the revised manuscript (Results session, page 10, line 401) while the interaction maps are presented in the supplementary material (Supplementary File – Figures S1-S5, pages S2 and S3).
About the requested in vitro binding, firstly, we would like to emphasize that we deeply understand the reviewer's concern regarding the experimental binding assay. We recognize that this assay is crucial for validating the results of molecular docking. However, now, we cannot conduct this analysis in our laboratory, and the partner laboratory responsible for the molecular docking analysis also lacks the means to perform it. For this reason, we acknowledge this as a limitation of our study. Despite this limitation, the molecular docking assay is widely employed, producing reliable results. In the present study, we have obtained a comprehensive set of results, highlighting innovative and promising findings for this compound.
Round 3
Reviewer 3 Report
Comments and Suggestions for Authors
-